# Allosteric stabilization of calcium and phosphoinositide dual binding engages several synaptotagmins in fast exocytosis

Janus RL Kobbersmed[1,2†], Manon MM Berns[2,3†], Susanne Ditlevsen[1], Jakob B Sørensen[2], Alexander M Walter[2,3]*

[1]Department of Mathematical Sciences, University of Copenhagen, Copenhagen, Denmark; [2]Department of Neuroscience, University of Copenhagen, Copenhagen, Denmark; [3]Molecular and Theoretical Neuroscience, Leibniz-Institut für Molekulare Pharmakologie, FMP im CharitéCrossOver, Berlin, Germany

*For correspondence:
awalter@sund.ku.dk

†These authors contributed equally to this work

**Abstract** Synaptic communication relies on the fusion of synaptic vesicles with the plasma membrane, which leads to neurotransmitter release. This exocytosis is triggered by brief and local elevations of intracellular $Ca^{2+}$ with remarkably high sensitivity. How this is molecularly achieved is unknown. While synaptotagmins confer the $Ca^{2+}$ sensitivity of neurotransmitter exocytosis, biochemical measurements reported $Ca^{2+}$ affinities too low to account for synaptic function. However, synaptotagmin's $Ca^{2+}$ affinity increases upon binding the plasma membrane phospholipid $PI(4,5)P_2$ and, vice versa, $Ca^{2+}$ binding increases synaptotagmin's $PI(4,5)P_2$ affinity, indicating a stabilization of the $Ca^{2+}/PI(4,5)P_2$ dual-bound state. Here, we devise a molecular exocytosis model based on this positive allosteric stabilization and the assumptions that (1.) synaptotagmin $Ca^{2+}/PI(4,5)P_2$ dual binding lowers the energy barrier for vesicle fusion and that (2.) the effect of multiple synaptotagmins on the energy barrier is additive. The model, which relies on biochemically measured $Ca^{2+}/PI(4,5)P_2$ affinities and protein copy numbers, reproduced the steep $Ca^{2+}$ dependency of neurotransmitter release. Our results indicate that each synaptotagmin engaging in $Ca^{2+}/PI(4,5)P_2$ dual-binding lowers the energy barrier for vesicle fusion by ~5 $k_BT$ and that allosteric stabilization of this state enables the synchronized engagement of several (typically three) synaptotagmins for fast exocytosis. Furthermore, we show that mutations altering synaptotagmin's allosteric properties may show dominant-negative effects, even though synaptotagmins act independently on the energy barrier, and that dynamic changes of local $PI(4,5)P_2$ (e.g. upon vesicle movement) dramatically impact synaptic responses. We conclude that allosterically stabilized $Ca^{2+}/PI(4,5)P_2$ dual binding enables synaptotagmins to exert their coordinated function in neurotransmission.

## Editor's evaluation

The calcium dependence of vesicle exocytosis at synapses is a power law with an exponent n = 3 or 4, however, the molecular mechanisms that underpin this highly non-linear dependence on calcium are unclear. To shed light on this fundamental question the authors build a model where 2 calcium ions bind to the protein synaptotagmin and synaptotagmin binds to the negatively charged lipid PIP2 in the presynaptic membrane. Simulations fit best the data from the calyx of Held synapse when 3 synaptotagmin molecules each bind calcium and PIP2. This compelling model shows that each Ca-synaptotagmin-PIP2 complex reduces the energy barrier for vesicle fusion by ~5k, thus, fast exocytosis at CNS synapses may require only 3 Ca-synaptogamin-PIP2 molecules to achieve submillisecond speeds of vesicle fusion.

**eLife digest** For our brains and nervous systems to work properly, the nerve cells within them must be able to 'talk' to each other. They do this by releasing chemical signals called neurotransmitters which other cells can detect and respond to.

Neurotransmitters are packaged in tiny membrane-bound spheres called vesicles. When a cell of the nervous system needs to send a signal to its neighbours, the vesicles fuse with the outer membrane of the cell, discharging their chemical contents for other cells to detect. The initial trigger for neurotransmitter release is a short, fast increase in the amount of calcium ions inside the signalling cell. One of the main proteins that helps regulate this process is synaptotagmin which binds to calcium and gives vesicles the signal to start unloading their chemicals.

Despite acting as a calcium sensor, synaptotagmin actually has a very low affinity for calcium ions by itself, meaning that it would not be efficient for the protein to respond alone. Synpatotagmin is more likely to bind to calcium if it is attached to a molecule called $PIP_2$, which is found in the membranes of cells The effect also occurs in reverse, as the binding of calcium to synaptotagmin increases the protein's affinity for $PIP_2$. However, how these three molecules – synaptotagmin, $PIP_2$, and calcium – work together to achieve the physiological release of neurotransmitters is poorly understood.

To help answer this question, Kobbersmed, Berns et al. set up a computer simulation of 'virtual vesicles' using available experimental data on synaptotagmin's affinity with calcium and $PIP_2$. In this simulation, synaptotagmin could only trigger the release of neurotransmitters when bound to both calcium and $PIP_2$. The model also showed that each 'complex' of synaptotagmin/calcium/$PIP_2$ made the vesicles more likely to fuse with the outer membrane of the cell – to the extent that only a handful of synaptotagmin molecules were needed to start neurotransmitter release from a single vesicle.

These results shed new light on a biological process central to the way nerve cells communicate with each other. In the future, Kobbersmed, Berns et al. hope that this insight will help us to understand the cause of diseases where communication in the nervous system is impaired.

## Introduction

Regulated neurotransmitter (NT) release from presynaptic terminals is crucial for information transfer across chemical synapses. NT release is triggered by action potentials (APs), which are transient de- and repolarizations of the presynaptic membrane potential that induce $Ca^{2+}$ influx through voltage-gated channels. The resulting brief and local elevations of the intracellular $Ca^{2+}$ concentration ($[Ca^{2+}]_i$) trigger the fusion of NT-containing synaptic vesicles (SVs) from the so-called readily releasable pool (RRP), whose SVs are localized (docked) at the plasma membrane and molecularly matured (primed) for fusion (*Kaeser and Regehr, 2017*; *Südhof, 2013*; *Verhage and Sørensen, 2008*). A high $Ca^{2+}$ sensitivity of NT release is needed to achieve fast responses to the very short AP-induced $Ca^{2+}$ transient and correspondingly, the SV fusion rate depends to the 4th-5th power on the $[Ca^{2+}]_i$ (*Bollmann et al., 2000*; *Burgalossi et al., 2010*; *Heidelberger et al., 1994*; *Schneggenburger and Neher, 2000*). Accordingly, previous models of NT release have assumed the successive binding of five $Ca^{2+}$ ions to a sensor that regulates release (*Bollmann et al., 2000*; *Lou et al., 2005*; *Schneggenburger and Neher, 2000*). However, how these macroscopic properties arise from the molecular components involved in SV fusion is still unknown.

The energy for SV fusion is provided by the assembly of the neuronal SNARE complex, which consists of vesicular synaptobrevin/VAMP and plasma membrane bound SNAP25 and syntaxin proteins (*Jahn and Fasshauer, 2012*; *Südhof, 2013*). $Ca^{2+}$ sensitivity of SV fusion is conferred by the vesicular protein synaptotagmin (syt), which interacts with the SNAREs (*Brewer et al., 2015*; *Littleton et al., 1993*; *Mohrmann et al., 2013*; *Schupp et al., 2017*; *Zhou et al., 2015*; *Zhou et al., 2017*). Several syt isoforms are expressed in synapses. Depending on the synapse type (e.g. mouse hippocampal pyramidal neurons or the Calyx of Held), syt1 or syt2 is required for synchronous, $Ca^{2+}$-induced fusion (*Geppert et al., 1994*; *Kochubey et al., 2016*; *Kochubey and Schneggenburger, 2011*; *Südhof, 2013*). These two syt isoforms are highly homologous and contain two cytosolic, $Ca^{2+}$-binding domains, C2A and C2B (*Südhof, 2002*), of which the C2B domain has been shown to be essential, and in some cases even sufficient, for synchronous NT release (*Bacaj et al., 2013*; *Gruget et al., 2020*; *Kochubey and Schneggenburger, 2011*; *Lee et al., 2013*; *Mackler et al., 2002*). The

C2B domain contains two $Ca^{2+}$ binding sites on its top loops (*Fernandez et al., 2001*). In addition, a second binding site allows the C2B domain to bind to the signaling lipid phosphatidylinositol 4,5-bisphosphate ($PI(4,5)P_2$) located in the plasma membrane (*Bai et al., 2004*; *Fernández-Chacón et al., 2001*; *Honigmann et al., 2013*; *Li et al., 2006*; *Xue et al., 2008*), but might also participate in (possibly transient) SNARE interactions (*Brewer et al., 2015*; *Zhou et al., 2015*; *Zhou et al., 2017*). A third site, located in the far end of the C2B domain (R398 and R399 in mouse syt1), is also involved in both SNARE- and membrane contacts (*Nyenhuis et al., 2021*; *Xue et al., 2008*; *Zhou et al., 2015*). Via these interactions, the syt C2B domain can induce close membrane-membrane contact in vitro (*Araç et al., 2006*; *Chang et al., 2018*; *Honigmann et al., 2013*; *Nyenhuis et al., 2021*; *Seven et al., 2013*; *Xue et al., 2008*), stable vesicle-membrane docking (*Chang et al., 2018*; *Chen et al., 2021*; *de Wit et al., 2009*), as well as dynamic vesicle-membrane association upon $Ca^{2+}$ influx into the cell (*Chang et al., 2018*).

Despite its central role as the $Ca^{2+}$ sensor for NT release, the intrinsic $Ca^{2+}$ affinity of the isolated syt C2B domain is remarkably low ($K_D \approx 200\ \mu M$, *Radhakrishnan et al., 2009*; *van den Bogaart et al., 2012*), much lower than the $Ca^{2+}$ sensitivity of NT release (*Bollmann et al., 2000*; *Schneggenburger and Neher, 2000*). However, binding of the C2B domain to $PI(4,5)P_2$, which is enriched at synapses (*van den Bogaart et al., 2011a*), drastically increases its $Ca^{2+}$ affinity (*van den Bogaart et al., 2012*). Similarly, the affinity for $PI(4,5)P_2$ increases upon $Ca^{2+}$ binding (*Pérez-Lara et al., 2016*; *van den Bogaart et al., 2012*). This indicates a positive allosteric coupling between the binding sites for $Ca^{2+}$ and $PI(4,5)P_2$, which promotes dual binding of $Ca^{2+}/PI(4,5)P_2$ (*Li et al., 2006*; *Radhakrishnan et al., 2009*; *van den Bogaart et al., 2012*). As binding of both molecules to syt is involved in fusion (*Kedar et al., 2015*; *Li et al., 2006*; *Mackler et al., 2002*; *Mackler and Reist, 2001*; *Wang et al., 2016*; *Wu et al., 2021a*; *Wu et al., 2021b*), this positive allosteric coupling might be central to syt's function in triggering $Ca^{2+}$-induced exocytosis (*van den Bogaart et al., 2012*).

In this paper, we developed a mathematical model in which the dual binding of the C2B domain to $Ca^{2+}$ and $PI(4,5)P_2$ promotes fusion. The model, which is based on the measured affinities and allostericity of $Ca^{2+}$ and $PI(4,5)P_2$ binding, describes stochastic binding/unbinding reactions at the level of individual syts and stochastic SV fusion events. The model predicts that each C2B domain engaging in dual $Ca^{2+}/PI(4,5)P_2$ binding lowers the energy barrier for fusion by $\sim 5\ k_B T$. Our results indicate that during fast NT release most fusion events occur once three syts per SV simultaneously engage their C2B domains in dual $Ca^{2+}/PI(4,5)P_2$ binding. This simultaneous engagement of multiple syts crucially relies on the positive allosteric coupling between $Ca^{2+}$ and $PI(4,5)P_2$ binding. We explored consequences of putative mutations affecting $Ca^{2+}/PI(4,5)P_2$ binding and/or the allosteric coupling between the binding sites of both species and suggest that changes of the allostericity contribute to dominant negative effects. Moreover, dynamic changes of $PI(4,5)P_2$ accessibility for the syts (e.g. induced by SV movement to the plasma membrane) are predicted to dramatically impact synaptic responses. We conclude that allosterically stabilized $Ca^{2+}/PI(4,5)P_2$ dual binding to the C2B domain forms the molecular basis for synaptotagmins to exert their cooperative control of neurotransmitter release.

## Results

### An experiment-based model of the triggering mechanism for SV fusion based on molecular interactions between syt, $Ca^{2+}$, and $PI(4,5)P_2$

To develop an experiment-based model of NT release based on molecular properties of syt, we first described the reaction scheme of a single C2B domain. The C2B domain binds $PI(4,5)P_2$ and two $Ca^{2+}$ ions (*Fernández-Chacón et al., 2002*; *Honigmann et al., 2013*; *Mackler et al., 2002*; *van den Bogaart et al., 2012*; *Xue et al., 2008*). We assumed the simplest case of simultaneous association of both $Ca^{2+}$ ions to the syt1 C2B domain. Therefore, in our model the C2B domain can be in four different states (*Figure 1A*): (1) an 'unbound' state, (2) a $PI(4,5)P_2$-bound state, (3) a state with two $Ca^{2+}$ ions bound, and (4) a 'dual-bound' state in which the C2B simultaneously engages $Ca^{2+}/PI(4,5)P_2$ binding. The affinities for $Ca^{2+}$ and $PI(4,5)P_2$ were set to those measured in vitro ($K_{D,2Ca2+}$ and $K_{D,PIP2}$)(*van den Bogaart et al., 2012*). Binding of $PI(4,5)P_2$ to the C2B domain was shown to increase the domain's affinity for $Ca^{2+}$ and vice versa, indicating a positive allosteric coupling between the two binding sites (*van den Bogaart et al., 2012*). We therefore implemented a positive allosteric stabilization of the dual-bound state in the model (illustrated by the red shaded areas of the C2B domain in *Figure 1A*)

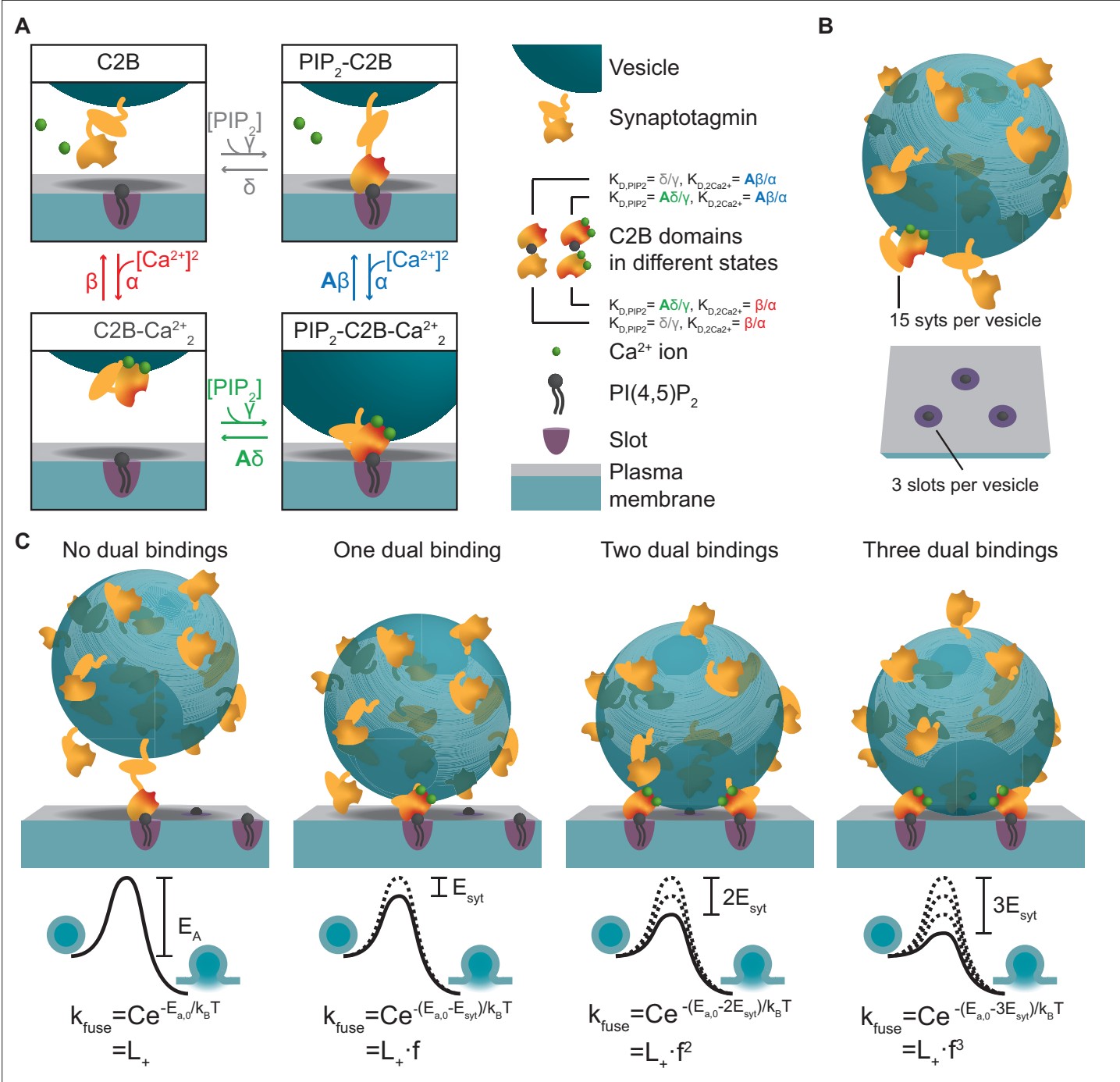

**Figure 1.** A molecular model of NT release triggered by $Ca^{2+}$ and $PI(4,5)P_2$ binding to the syt1 C2B domain. (**A**) The reaction scheme of a single syt C2B domain. Each syt can be in one of four binding states: Nothing bound (top left), $PI(4,5)P_2$ bound (top right), two $Ca^{2+}$ ions bound (bottom left), and $PI(4,5)P_2$ and two $Ca^{2+}$ ions bound (bottom right). Simultaneous binding of $Ca^{2+}$ and $PI(4,5)P_2$ to the syt C2B domain is referred to as dual binding. The factor A<1 on the dissociation rates ($\beta$ and $\delta$) from the dual-bound state represents the positive allosteric effect of simultaneous $PI(4,5)P_2$ and $Ca^{2+}$ binding and leads to stabilization of the dual-bound state. The ratio between dissociation rate and association rate constants ($\beta/\alpha$ and $\delta/\gamma$) is equal to the respective dissociation constants of syt1 determined in vitro ($K_{D,2Ca2+} = 221^2\ \mu M^2$ and $K_{D,PIP2} = 20\ \mu M$, *van den Bogaart et al., 2012*). An alternative reaction scheme where $Ca^{2+}$ binding leads to association of the C2B domain with the plasma membrane is shown in *Figure 1—figure supplement 1*. Our model is not influenced by the assumptions on whether $Ca^{2+}$ binding to syt leads to plasma membrane or vesicle association. (**B**) The stoichiometry at the SV fusion site. We assume 15 syts per SV (*Takamori et al., 2006*), and that the association of the syt C2B domain to $PI(4,5)P_2$ is limited to a finite number of slots (here illustrated for $M_{slots} = 3$). (**C**) The effect of formation of multiple dual bindings on the energy barrier for SV fusion and the SV fusion rate. We assume that each dual-binding C2B domain lowers the energy barrier for fusion by the same amount ($E_{syt}$, illustrated in middle row), thereby increasing the

*Figure 1 continued on next page*

*Figure 1 continued*

fusion rate ($k_{fuse}$) with a factor *f* for each dual binding (equation in bottom row). The fusion rate for an SV with no dual bindings formed is set to $L_+$. The model is a Markov model, which can be summarized in a state diagram describing the reactions of the syt-harboring SV (***Figure 1—figure supplement 2***).

The online version of this article includes the following figure supplement(s) for figure 1:

**Figure supplement 1.** Alternative reaction scheme of a single syt in which $Ca^{2+}$ binding leads to association to the plasma membrane.

**Figure supplement 2.** Reaction scheme for all reactions of an entire SV.

by introducing the allosteric factor (A=0.00022, see Materials and methods ***van den Bogaart et al., 2012***) which slows down the $Ca^{2+}$ and PI(4,5)$P_2$ dissociation from the dual-bound state.

We next extended the model to the level of the complete SV. On average, SVs contain 15 copies of syt1 (***Figure 1B***; ***Takamori et al., 2006***), which may work together to regulate SV fusion. Spontaneous release occurs at low rates (with a rate constant '$L_+$'), reflected by a high initial energy barrier for SV fusion (***Figure 1C***). Because syt's stimulation of SV fusion likely relies on the simultaneous binding of both $Ca^{2+}$ and PI(4,5)$P_2$ (***Kedar et al., 2015***; ***Li et al., 2006***; ***Mackler et al., 2002***; ***Mackler and Reist, 2001***; ***Wang et al., 2016***; ***Wu et al., 2021a***; ***Wu et al., 2021b***), we assumed that each dual-bound C2B domain promotes exocytosis by lowering this barrier. How this might be achieved exactly is unknown, but could involve bridging plasma and SV membranes (***Figure 1A***), changing the curvature of the plasma membrane (***Figure 1—figure supplement 1***), changing the local electrostatic environment, or directly or indirectly promoting SNARE complex assembly (***Bhalla et al., 2006***; ***Martens et al., 2007***; ***Ruiter et al., 2019***; ***Schupp et al., 2017***; ***Tang et al., 2006***; ***van den Bogaart et al., 2011b***; ***Zhou et al., 2015***; ***Zhou et al., 2017***). We assumed that multiple syts may progressively lower the energy barrier by the successive engagement of their C2B domains in dual $Ca^{2+}$/PI(4,5)$P_2$ binding, and investigated the simplest scenario, in which each dual-bound C2B domain contributed the same amount of energy ($E_{syt}$) thereby increasing the fusion rate by the same factor (*f*) (***Figure 1C***; ***Schotten et al., 2015***). The number of syts that can simultaneously promote fusion may be limited by their access to PI(4,5)$P_2$ in the plasma membrane. This could be due to limited space beneath the SV or limited molecular access to bind PI(4,5)$P_2$ clusters and/or SNAREs (***de Wit et al., 2009***; ***Mohrmann et al., 2013***; ***Rickman and Davletov, 2003***). We therefore created a model that stochastically describes the binding status of individual SVs (***Figure 1—figure supplement 2***) with a limited number of PI(4,5)$P_2$ association possibilities ('slots') and investigated how this number affects physiological responses.

## At least three PI(4,5)$P_2$ binding slots are required to reproduce release kinetics from the calyx of Held synapse

A hallmark of the NT release reaction is its large dynamic range in response to $Ca^{2+}$ stimuli as impressively demonstrated by experimental data from the calyx of Held synapse where release latencies (defined as the time of the fifth SV fusion after the stimulus) and exocytosis rates have been measured for a broad range of $Ca^{2+}$ concentrations using $Ca^{2+}$ uncaging (***Bollmann et al., 2000***; ***Kochubey and Schneggenburger, 2011***; ***Lou et al., 2005***; ***Schneggenburger and Neher, 2000***; ***Sun et al., 2007***). At this well-established model synapse, fast NT release is controlled by syt2, which is functionally redundant with syt1 in neurons (***Kochubey et al., 2016***; ***Xu et al., 2007***).

We evaluated whether our model could reproduce this $Ca^{2+}$ dependence by simulating release latencies and peak release rates in response to step-like $Ca^{2+}$ stimuli. The ability to reproduce the experimental data depended on the number of 'slots' for syt PI(4,5)$P_2$ binding (***Figure 2A***). We first fitted the free parameters in our model by optimizing the agreement (i.e. by reducing a pre-defined cost function, see Materials and methods) between model predictions and release rates and latencies determined experimentally by Kochubey and Schneggenburger (***Figure 2A***; ***Kochubey and Schneggenburger, 2011***). During this fitting process, we took the entire distribution of the experimentally obtained release latencies into account by using the likelihood function (see Materials and methods). This was not feasible for the experimental peak release rates (since accurate computation of the maximum rate of stochastic events is not feasible), which were therefore compared to the closed form solution of the model (see Materials and methods). Because the affinities for $Ca^{2+}$ and PI(4,5)$P_2$, and the allosteric coupling between both species ($K_{D,2Ca2+}$, $K_{D,PIP2}$ and *A*, ***Figure 1A***, ***Figure 1—figure supplement 2***) as well as the RRP size (including its variance) were taken from literature (***Table 1***;

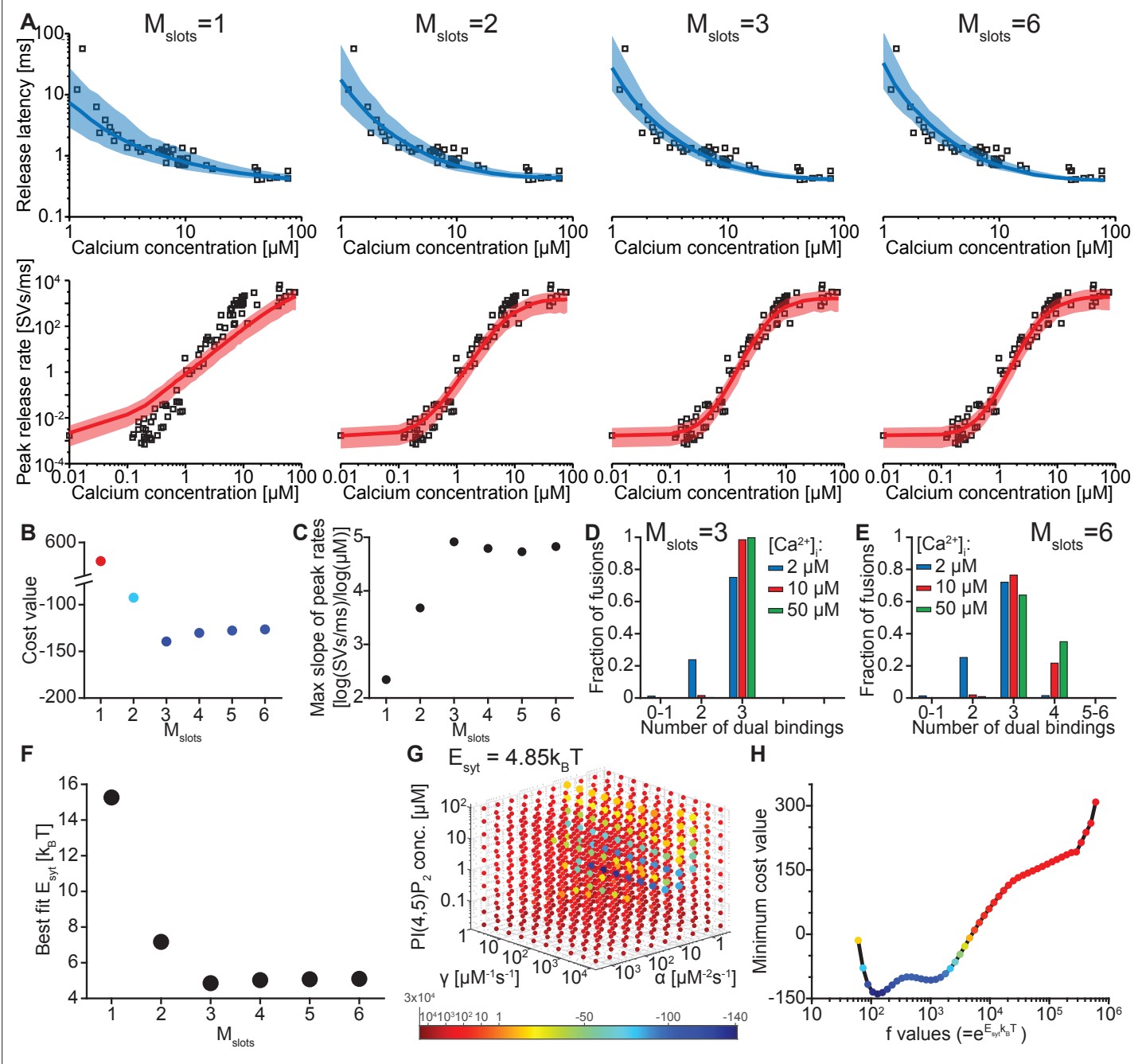

**Figure 2.** The model reproduces the Ca²⁺ dependency of SV fusion when at least three syts can simultaneously bind PI(4,5)P₂. (**A**) Best fit results for different choices of $M_{slots}$. The top panels show best fit model prediction of the release latencies (time to fifth SV fusion), and the bottom panels show the predicted peak release rates at varying Ca²⁺ concentrations. The black points are experimental data (individual measurements replotted from *Kochubey and Schneggenburger, 2011*). Solid lines represent the median release latencies and mean peak release rates predicted by the model from 1000 repetitions per simulated [Ca²⁺]ᵢ. The shaded areas indicate the 95% prediction interval of the model. The models with $M_{slots}$ <3 failed to reproduce data, whereas models with $M_{slots}$ ≥3 agreed with data. Optimization and simulation were performed using a variable RRP size (*Figure 2—figure supplement 1*). (**B**) Minimum cost value as a function of $M_{slots}$. With $M_{slots}$ = 3 the minimum cost value was obtained, indicating the best correspondence to experimental release latencies and peak release rates. The point colors correspond to the color scale in panel G. (**C**) Maximal slope of logarithm of simulated peak release rate vs logarithm of [Ca²⁺]ᵢ. For $M_{slots}$ <3 the model resulted in a too shallow Ca²⁺ dependency of release rates. (**D–E**) The number of dual bindings at the time point of fusion for $M_{slots}$ = 3 (**D**) and $M_{slots}$ = 6 (**E**) determined from simulations of 10⁴ SVs using three different [Ca²⁺]ᵢ. Most fusions took place after forming three bindings, even when allowing more dual bindings to form. A larger set of Ca²⁺ stimuli was also explored (*Figure 2—figure supplement 2* and *Figure 2—figure supplement 3*). Forcing a model with M_slots = 6 to fuse after 5/6 dual bindings were formed could not describe the experimental data and showed a too steep Ca²⁺ dependency of release (*Figure 2—figure supplement 4*). The number of

*Figure 2 continued on next page*

*Figure 2 continued*

dual bounds before fusion was slightly sensitive to the assumption on how many $Ca^{2+}$ ions bind to a single syt (*Figure 2—figure supplement 5*). This simplified model assuming simultaneous binding of two $Ca^{2+}$-ions agrees with one where the binding is sequential if the binding of the second $Ca^{2+}$ ion is favoured (*Figure 2—figure supplement 6*) (**F**) The change in the energy barrier induced by dual binding formation ($E_{syt}$) as a function of $M_{slots}$. $E_{syt}$ was computed from the fitted $f$ values and was approximately constant for $M_{slots} \geq 3$. (**G**) Exploring cost values in the parameter space for a model with $M_{slots} = 3$. With $f$ fixed at the best fit value ($f=128$), we determined the cost value of all combinations of 30 choices of the three free parameters, $\alpha$, $\gamma$ and [PI(4,5)P$_2$]. As the added delay only leads to a vertical shift in the release latencies plot (see *Figure 2—figure supplement 7*), this parameter was optimized for each choice of the other free parameters to minimize the costs. The plot shows a subset of the parameter combinations, and the colors indicate the cost value at each point. The color scale is linear below 1 and logarithmic above 1, and points with a cost value > 1 are smaller for better visibility. The darkest blue colored ball represents the overall minimum cost value in this parameter search and agrees with the best fit obtained. The effect of varying each of the free parameters on release latencies and peak release rates can be seen in *Figure 2—figure supplement 7*. (**H**) Minimum cost value as a function of $f$ for a model with $M_{slots} = 3$. For each choice of $f$ the model was fit to experimental data. This parameter exploration found the same minimum in the parameter space as found by fitting all free parameters. Simulation scripts can be found in *Source code 1*. Results depicted in *Figure 2* and its figure supplements can be found in *Figure 2—source data 1*.

The online version of this article includes the following source data and figure supplement(s) for figure 2:

**Source data 1.** Source data for *Figure 2* and *Figure 2—figure supplements 2–7*.

**Figure supplement 1.** RRP distribution.

**Figure supplement 2.** Exploration of the number of dual bindings formed before fusion of an SV with M$_{slots}$ = 3.

**Figure supplement 3.** Exploration of the number of dual bindings formed before fusion of an SV with M$_{slots}$ = 6.

**Figure supplement 4.** Forcing fusion from a state in which 5–6 syts are dual binding $Ca^{2+}$ and PI(4,5)P$_2$ causes a too steep $Ca^{2+}$ dependency of the peak release rates.

**Figure supplement 5.** Exploration of how the estimated number of dual binding syts for fusion depends on the number of $Ca^{2+}$ ions bound to one C2B domain.

**Figure supplement 6.** A model with two sequential $Ca^{2+}$ binding steps compared to the simplified model with the simultaneous binding of two $Ca^{2+}$ ions.

**Figure supplement 7.** Effect of the free parameters on release latencies and peak release rates.

---

*Figure 2—figure supplement 1*; *van den Bogaart et al., 2012*; *Wölfel and Schneggenburger, 2003*), we only had to estimate five parameters: (1) the binding rate constant of the two $Ca^{2+}$ ions ($\alpha$), (2) the binding rate constant of PI(4,5)P$_2$ ($\gamma$), (3) the PI(4,5)P$_2$ concentration ([PI(4,5)P$_2$]), (4) the reduction of the energy barrier for fusion induced by $Ca^{2+}$/PI(4,5)P$_2$ dual binding of one C2B domain ($E_{syt}$) and (5) a fixed delay ($d$) between time of uncaging and response onset, like in previous models (see *Kochubey and Schneggenburger, 2011*; *Schneggenburger and Neher, 2000*). Having obtained the best fit parameters, peak release rates and release latencies (*Figure 2A*) were stochastically simulated based on the closed-form solution of the Markov model (see Materials and methods).

---

**Table 1.** Fixed parameters in the model.

| Parameter | Description | Value | Reference |
|---|---|---|---|
| $n_{syts}$ | Number of syts per SV | 15 | *Takamori et al., 2006* |
| $M_{slots}$ | Number of binding slots for syts to PI(4,5)P$_2$ (see *Figure 2*) | Varied from 1 to 6 | This paper |
| $n_{ves}$ | Number of RRP vesicles | Mean: 4000, sd: 2000, gamma distribution | *Wölfel and Schneggenburger, 2003* |
| $[Ca^{2+}]_0$ | Resting $[Ca^{2+}]_i$ | 0.05 μM | *Helmchen et al., 1997* |
| $K_{D,2Ca2+}$ | Dissociation constant of C2B for two $Ca^{2+}$ ions | $221^2$ μM$^2$ | *van den Bogaart et al., 2012* |
| $K_{D,PIP2}$ | Dissociation constant of C2B for PI(4,5)P$_2$ | 20 μM | *van den Bogaart et al., 2012* |
| $A$ | Allosteric factor see Materials and methods for calculation | 0.00022 | *van den Bogaart et al., 2012* |
| $\beta$ | $Ca^{2+}$ unbinding rate constant | $K_{D,2Ca2+} \cdot \alpha$ | Computed using best fit $\alpha$ (see *Table 2*) |
| $\delta$ | PI(4,5)P$_2$ unbinding rate constant of C2B | $K_{D,PIP2} \cdot \gamma$ | Computed using best fit $\alpha$ (see *Table 2*) |
| $L_+$ | Basal fusion rate | $4.23 \cdot 10^{-4}$ s$^{-1}$ | Computed using data from *Kochubey and Schneggenburger, 2011* |

**Table 2.** Best fit model parameters and corresponding costs with different number of slots.

| Parameter | Description | $M_{slots}$ = 1 | $M_{slots}$ = 2 | $M_{slots}$ = 3 | $M_{slots}$ = 4 | $M_{slots}$ = 5 | $M_{slots}$ = 6 |
|---|---|---|---|---|---|---|---|
| $\alpha$ ($\mu M^{-2}s^{-1}$) | $Ca^{2+}$ association rate constant | 0.03712 | 34.99 | 24.70 | 25.08 | 24.51 | 24.11 |
| $\gamma$ ($\mu M^{-1}s^{-1}$) | $PI(4,5)P_2$ association rate constant | $1.425 \cdot 10^5$ | 572.6 | 124.7 | 121.3 | 124.31 | 126.6 |
| [$PI(4,5)P_2$] ($\mu M$) | Effective $PI(4,5)P_2$ concentration for syt | 0.009658 | 0.2523 | 1.109 | 0.4528 | 0.3048 | 0.2320 |
| $f$<br>($E_{syt}$ ($k_bT$)) | Factor on the release rate for each $Ca^{2+}/PI(4,5)P_2$ dual bound C2B domain (resulting from a fusion barrier reduction of $E_{syt}$) | $4.259 \cdot 10^6$ (15.3) | 1298 (7.17) | 128.2 (4.85) | 152.1 (5.02) | 159.6 (5.07) | 163.5 (5.10) |
| $d$ (ms) | Added delay | 0.3211 | 0.3761 | 0.3803 | 0.3866 | 0.3876 | 0.3881 |
| Costs | Quantification of goodness of fit | 581.9 | −92.50 | −139.4 | −130.3 | −127.7 | −126.5 |

We systematically varied the number of slots ($M_{slots}$) from one to six, optimized the free parameters for each of these choices, and compared the best fit solutions. With a single slot ($M_{slots}$ = 1), the model accounted for experimentally observed release latencies, but failed to reproduce the steep relationship between [$Ca^{2+}$]$_i$ and peak release rates (*Figure 2A*). Adding more slots strongly improved the agreement with experimental data. The best agreement was found with three slots ($M_{slots}$ = 3) and the agreement slightly decreased with more slots ($M_{slots}$ >3) (*Figure 2B*). However, all models with at least three slots ($M_{slots}$ ≥3) captured the steep dependency of peak release rates on [$Ca^{2+}$]$_i$, with a maximum slope of 4–5 on a double-logarithmic plot (*Figure 2C*; *Schneggenburger and Neher, 2000*).

Our model made it possible to inspect the fate of each individual fusing SV, including the number of synaptotagmins dually binding $Ca^{2+}$ and $PI(4,5)P_2$ just before fusion. Remarkably, even in models with more than three slots ($M_{slots}$ >3), fast NT release, induced by moderate to high [$Ca^{2+}$]$_i$, was predicted to primarily engage three dual-bound syt C2B domains. At lower [$Ca^{2+}$]$_i$ fewer dual-bound syts mediated fusion (*Figure 2D–E*, *Figure 2—figure supplement 2*, *Figure 2—figure supplement 3*). Correspondingly, the estimated reduction of the energy barrier for fusion by each $Ca^{2+}/PI(4,5)P_2$ dual binding C2B domain was similar ($E_{syt} \approx 5\ k_BT$) for all versions of the model with at least three slots ($M_{slots}$ ≥3) (*Figure 2F*, see *Table 2* for best fit model parameters for each setting of $M_{slots}$). Our model thus predicts that most fusion events occur when three syts actively reduce the energy barrier for fast SV fusion.

We next investigated to which extent the estimated number of syts working together for fusion depended on some of the assumptions of our model. For instance, if each syt dual-binding $Ca^{2+}/PI(4,5)$ $P_2$ had a lower effect on the vesicle fusion rate, might this be compensated by more syts working together during fusion? We investigated this by manually forcing a lower individual contribution to the energy barrier for fusion in a model with six slots and refitting the remaining parameters. Under such conditions, the dependence of the peak release rate on [$Ca^{2+}$]$_i$ became too steep, indicating that too many syts working together make the $Ca^{2+}$ sensitivity unnaturally high (*Figure 2—figure supplement 4*). We then investigated how the assumption of simultaneous binding of two $Ca^{2+}$ ions to the C2B domain affected the conclusions. If each C2B domain only bound a single $Ca^{2+}$ ion, the dependence of the peak release rate on [$Ca^{2+}$]$_i$ was too shallow, even in a model with six slots. Allowing the number of $Ca^{2+}$ ions binding to one C2B domain to vary in a macroscopic version of the model with six slots predicted the binding of 1.53 $Ca^{2+}$ ions per C2B domain and most NT release commencing with four or fewer cooperating syts (*Figure 2—figure supplement 5*). This confirms that most C2B domains need to bind two $Ca^{2+}$ ions to exert their effect. Simulating a model with consecutive $Ca^{2+}$ binding to the two binding sites of the C2B domain on all syts of RRP SVs would be computationally too costly (and would involve additional, unknown parameters). However, we show that our simplification of simultaneous binding aligns with such a more complex model if the binding of the second $Ca^{2+}$ ion is favored (*Figure 2—figure supplement 6*), which is reasonable based on the proximity to negatively charged lipid headgroups following the insertion of the C2B domain into the plasma membrane. Thus, while our model is a simplification of the reality, the main conclusion on the number of slots needed ($M_{slots}$ ≥3) and the number of syts sufficient to mediate fast NT release (≤4), are robust estimates. Because our molecular model assuming three slots ($M_{slots}$ = 3) and simultaneous binding of two $Ca^{2+}$

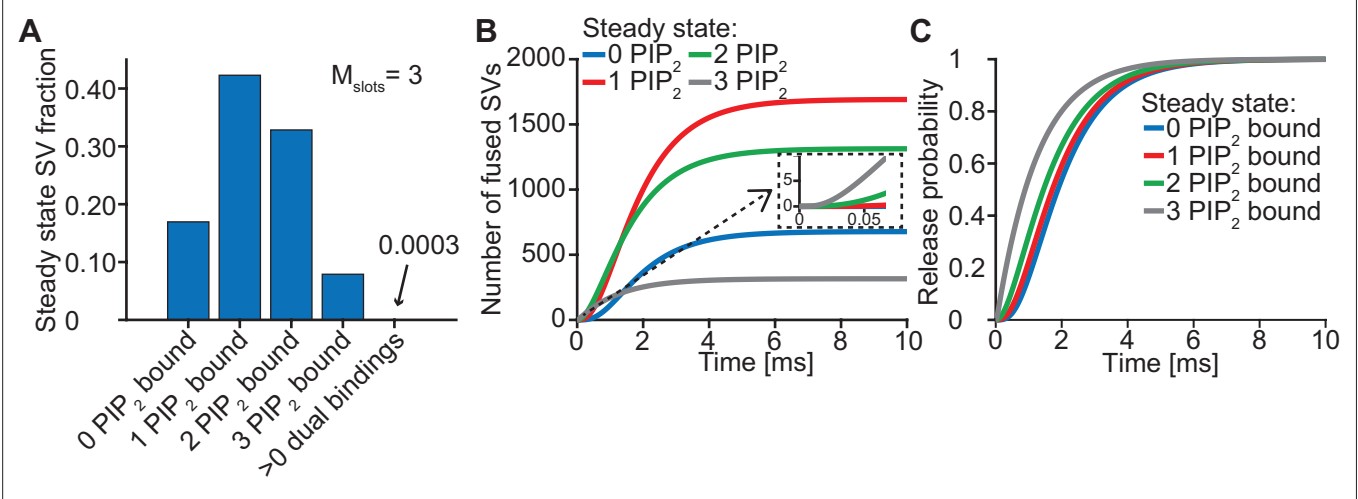

**Figure 3.** Syts binding to PI(4,5)P$_2$ prior to Ca$^{2+}$ stimulus underlies very fast SV fusion. (**A**) PI(4,5)P$_2$ binding status of SVs at steady state. At resting [Ca$^{2+}$]$_i$ of 50 nM, more than 40% of SVs have bound a single PI(4,5)P$_2$ molecule (not including those that have formed a dual binding), more than 30% have bound two PI(4,5)P$_2$, while less than 10% have bound three PI(4,5)P$_2$. Close to no SVs form dual bindings at steady state. (**B**) Cumulative fusion of SVs after 50 µM step Ca$^{2+}$ at t=0, grouped according to their initial PI(4,5)P$_2$ binding state. During the first ~0.5ms, release is dominated by SVs having two or three syts bound to PI(4,5)P$_2$ prior to the stimulus. The insert shows that the SVs having prebound three PI(4,5)P$_2$ constitute the majority of the first five SVs that fuse in response to the Ca$^{2+}$ step and therefore largely impact the release latency. (**C**) Cumulative release probability of SVs over time after 50 µM step Ca$^{2+}$ at t=0, grouped according to initial PI(4,5)P$_2$ binding state. The dominance of SVs having pre-bound to PI(4,5)P$_2$ with two or three syts in panel B is explained by their high release probability compared to SVs with no or only one PI(4,5)P$_2$ bound. *Figure 3—figure supplement 1* shows the same analysis for a model with M$_{slots}$ = 6. Simulation scripts can be found in *Source code 1*. Depicted simulation results can be found in *Figure 3— source data 1*.

The online version of this article includes the following source data and figure supplement(s) for figure 3:

**Source data 1.** Source data for *Figure 3* and *Figure 3—figure supplement 1*.

**Figure supplement 1.** Syts binding to PI(4,5)P$_2$ prior to the Ca$^{2+}$ stimulus underlies very fast SV fusion (model with M$_{slots}$ = 6).

ions to the syt C2B domains was the simplest model that accounted for the experimental data, we used it for further simulations.

The best fit parameters for three slots revealed rapid association rate constants for Ca$^{2+}$ and PI(4,5)P$_2$ to the C2B domain and PI(4,5)P$_2$ levels corresponding to a concentration of ~1 µM in an in vitro setting (*Table 2*). Predicted responses obtained using the best fit parameters were sensitive to changes of either of these parameters (*Figure 2—figure supplement 7*). For instance, higher levels of PI(4,5)P$_2$ decreased the release latencies and increased the rate of fusion, and changing the Ca$^{2+}$ association rate constant ($\alpha$) affected the release latencies much more than changing the PI(4,5)P$_2$ association rate constant ($\gamma$). We verified that these parameters represent unique solutions by systematically exploring the parameter space with f (which relates to the lowering of the fusion barrier for each syt dual-binding Ca$^{2+}$/PI(4,5)P$_2$) fixed to the best fit value (f=128), which revealed a clear minimum at the best fit parameters (*Figure 2G*, darkest ball). We furthermore confirmed that this f value was optimal by systematically varying f and fitting all other parameters (*Figure 2H*).

## The number of syt proteins pre-associated to PI(4,5)P$_2$ at rest influences the SV's Ca$^{2+}$ responsiveness

The steady state concentration of PI(4,5)P$_2$ determines the probability of syts associating to PI(4,5)P$_2$ at rest. With the best fit parameters, our model predicts that at rest ([Ca$^{2+}$]$_i$=50 nM) most SVs associate to PI(4,5)P$_2$ by engaging one (~42%), two (~33%) or three (~8%) syts (*Figure 3A*, see *Figure 3—figure supplement 1* for behavior in the model with M$_{slots}$ = 6). With a step-like Ca$^{2+}$ stimulus to 50 µM, SVs with two or three pre-associated syts mediated most of the fastest (<0.5ms) SV fusions (*Figure 3B*). Consequently, changing the steady state PI(4,5)P$_2$ concentration (which changes the number of pre-associated syts/SVs) largely impacted the release latencies (defined as the timing of the fifth SV that fuses, *Kochubey and Schneggenburger, 2011*; *Figure 2—figure supplement 7*). Due to the allosteric coupling, the Ca$^{2+}$ affinities of syts prebound to PI(4,5)P$_2$ are increased and SVs with more

PI(4,5)P$_2$ interactions are more responsive to the Ca$^{2+}$ stimulus (*Figure 3C*). Thus, at the single SV level, the number of pre-associated syts to PI(4,5)P$_2$ at rest plays a role in very fast (submillisecond) SV release and causes heterogeneity in release probability among RRP SVs.

## Allosteric stabilization of Ca$^{2+}$/PI(4,5)P$_2$ dual binding is necessary to synchronize multiple C2B domains for fast SV fusion

An important feature of our model is the inclusion of a positive allosteric interaction between Ca$^{2+}$ and PI(4,5)P$_2$ binding to the C2B domain which we based on increased affinities measured in vitro (*van den Bogaart et al., 2012*). To explore the physiological relevance of this allostericity, we investigated how individual SVs engaged their syt C2B domains in Ca$^{2+}$/PI(4,5)P$_2$ dual binding in response to a stepwise increase of [Ca$^{2+}$]$_i$ to 50 μM (*Figure 4A*) with or without this allosteric coupling. We did this by following the fate of the RRP SVs in stochastic simulations, four of which are illustrated in *Figure 4B*. Under normal conditions (with allostericity), syt C2B domains quickly associated both Ca$^{2+}$ and PI(4,5)P$_2$ and their respective allosteric stabilization slowed the dissociation of either species resulting in a lifetime of their dual binding of ~1.3 ms on average. This enabled the successive engagement of three dual-bound C2B domains for most RRP SVs (including all four illustrated SVs). The average waiting time for fusion for the RRP SVs was ~1.1 ms (*Figure 4B*, fusion indicated by circles). By inspecting the average behavior of the entire RRP of SVs it became clear that the overall release rate closely followed the population of SVs engaging three C2B domains in dual Ca$^{2+}$/PI(4,5)P$_2$ binding, illustrating the importance of engaging three syts for fast SV fusion in this model (*Figure 4C*). We also simulated the postsynaptic response produced by this NT release by convolving the SV release rate with a typical postsynaptic response to the fusion of a single SV (see Materials and methods), which revealed synchronous and large Excitatory Post Synaptic Currents (EPSCs)(*Figure 4D*).

We then explored what would happen without the allosteric stabilization of Ca$^{2+}$/PI(4,5)P$_2$ dual binding (by setting A=1; *Figure 4E*). In this case, the C2B domains still quickly associated Ca$^{2+}$ and PI(4,5)P$_2$, but without the allosteric slowing of Ca$^{2+}$/PI(4,5)P$_2$ dissociation the lifetime of dual-bound C2B domains was dramatically reduced to an average of ~0.0003 ms. This made it very improbable to engage multiple C2B domains in dual Ca$^{2+}$/PI(4,5)P$_2$ binding (*Figure 4E*). In turn, without the simultaneous engagement of multiple syts dual-binding Ca$^{2+}$/PI(4,5)P$_2$, NT release became very unlikely. In fact, none of the randomly chosen four RRP SVs fused within 4 ms (*Figure 4E*). Inspection of the average behavior of the entire RRP revealed that only few SVs engaged more than one syt C2B domain in dual Ca$^{2+}$/PI(4,5)P$_2$ binding, resulting in a very low fusion rate (*Figure 4F*). Correspondingly, postsynaptic EPSCs were severely disrupted, and most release events were ill-synchronized single SV fusion events (*Figure 4G*). It was furthermore not possible to fit a model without the allosteric stabilization to the experimental dataset (*Figure 4—figure supplement 1*). Thus, the positive allosteric coupling between Ca$^{2+}$ and PI(4,5)P$_2$ is fundamental for the syts to simultaneously and persistently engage multiple C2B domains per SV in Ca$^{2+}$/PI(4,5)P$_2$ dual binding.

## Many syts per SV speed up exocytosis by increasing the probability of Ca$^{2+}$/PI(4,5)P$_2$ dual binding

Our model suggests that only a few syts simultaneously binding Ca$^{2+}$ and PI(4,5)P$_2$ are required to promote fast SV fusion (*Figure 2*). Yet, a total of 15 copies are expressed per SV on average (*Takamori et al., 2006*), which raises the question why SVs carry such excess and whether and how the additional syt copies contribute to the characteristics of Ca$^{2+}$-induced synaptic transmission. To investigate this, we simulated Ca$^{2+}$ uncaging experiments with reduced numbers of syts per SV while keeping all other parameters in the model constant. With fewer syts, release latencies increased and peak release rates reduced. Defects were particularly prominent for reductions to less than three copies per SV (*Figure 5A*). Further exploration indicated that the responses slowed down upon reductions in syt copy number because it took SVs longer to simultaneously engage three C2B domains in dual Ca$^{2+}$/PI(4,5)P$_2$ binding and that fewer SVs reached this state (*Figure 5B*).

While Ca$^{2+}$ uncaging stimuli are exquisitely suited to map the full range of synaptic responses, synaptic transmission is physiologically triggered by APs that induce short-lived Ca$^{2+}$ transients. To stochastically predict responses to such time-varying Ca$^{2+}$ stimuli, we implemented our model using the Gillespie algorithm. After verifying that this model implementation agreed with the initial implementation (*Figure 5—figure supplement 1*), we simulated responses to a typical AP-induced Ca$^{2+}$

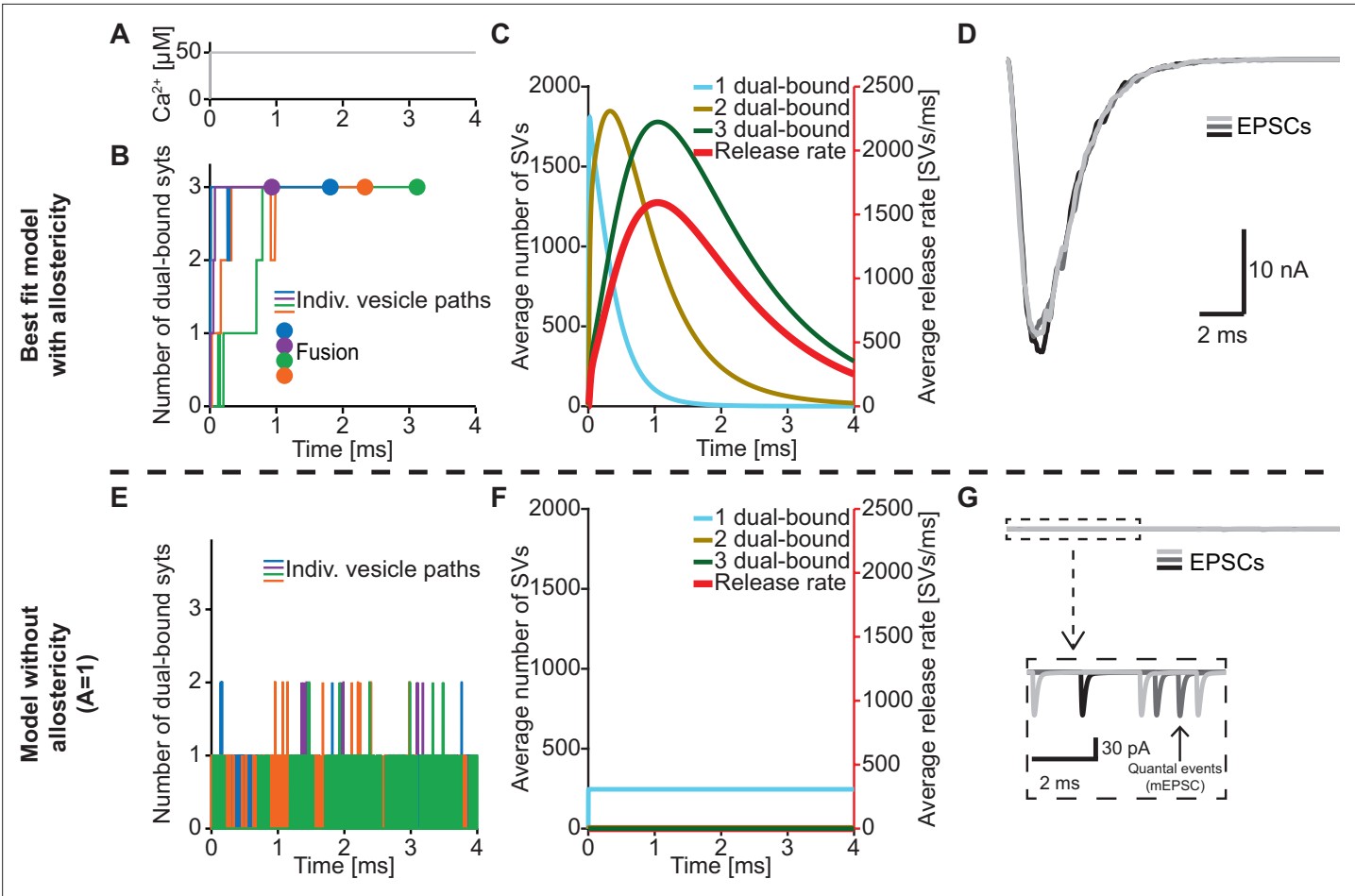

**Figure 4.** The positive allostericity between Ca$^{2+}$ and PI(4,5)P$_2$ allows multiple syt C2B domains to engage in Ca$^{2+}$/PI(4,5)P$_2$ dual binding. (**A**) Ca$^{2+}$ signal used in simulations ([Ca$^{2+}$]$_i$=50 μM). This constant Ca$^{2+}$ concentration was used for all simulations depicted in this figure. (**B**) The path towards fusion for four example SVs using stochastic simulations of the best fit model (with allostericity). The differently colored graphs show the number C2B domains engaging in Ca$^{2+}$/PI(4,5)P$_2$ dual binding for the four example SVs. The large dots indicate SV fusion. (**C**) Average number of SVs having one (blue), two (olive) and three (green) C2B domains engaging in Ca$^{2+}$/PI(4,5)P$_2$ dual binding and the fusion rate (red) over time in simulations including the entire RRP. In the best fit model, the number of SVs with three syts dual-binding Ca$^{2+}$/PI(4,5)P$_2$ peaks approximately at the same time as the fusion rate. The decrease in number of SVs with one or two C2B domains dual binding Ca$^{2+}$/PI(4,5)P$_2$ reflects formation of additional dual bindings. The decrease in total number of SVs is caused by fusion of RRP vesicles. (**D**) Excitatory Postsynaptic Currents (EPSCs) from three stochastic simulations with a fixed RRP size of 4000 SVs. The model predicts synchronous EPSCs with a small variation caused by the stochasticity of the molecular reactions. (**E**) The path towards fusion for four example SVs (similar to panel A) in the model without allostericity in stochastic simulation. All parameters other than the allosteric factor, $A$, are the same as in the best fit model. Without stabilization, the syts quickly engage *and* disengage in Ca$^{2+}$/PI(4,5)P$_2$ dual binding and rarely more than one syt engages in dual binding. Formation and dissociation of individual dual-bound syts is too fast to distinguish on the time scale depicted here. No fusions occurred in the depicted simulations. (**F**) Average number of SVs having one (blue), two (olive) and three (green) syts engaging in Ca$^{2+}$/PI(4,5)P$_2$ dual binding and the fusion rate (red) over time in the model without allostericity. Almost no SVs form more than one dual binding, which results in a very low fusion rate. (**G**) EPSCs from three stochastic simulations and with a fixed RRP size of 4000 SVs. A model lacking allostericity only shows sporadic, individual release events. The insert shows a zoom-in of the first 6ms of simulation and makes single SV fusion events giving rise to quantal, 'miniature' mEPSCs visible. Fitting the model without the allosteric effect to the experimental data was unsuccessful (*Figure 4—figure supplement 1*). Simulation scripts can be found in *Source code 1*. Results from fitting the model without allosteric effect can be found in .

The online version of this article includes the following source data and figure supplement(s) for figure 4:

**Figure supplement 1.** Fitting of the model without allosteric interaction between Ca$^{2+}$ and PI(4,5)P$_2$ fails to reproduce the Ca$^{2+}$ dependency of NT release.

**Figure supplement 1—source data 1.** Source data for *Figure 4—figure supplement 1*.

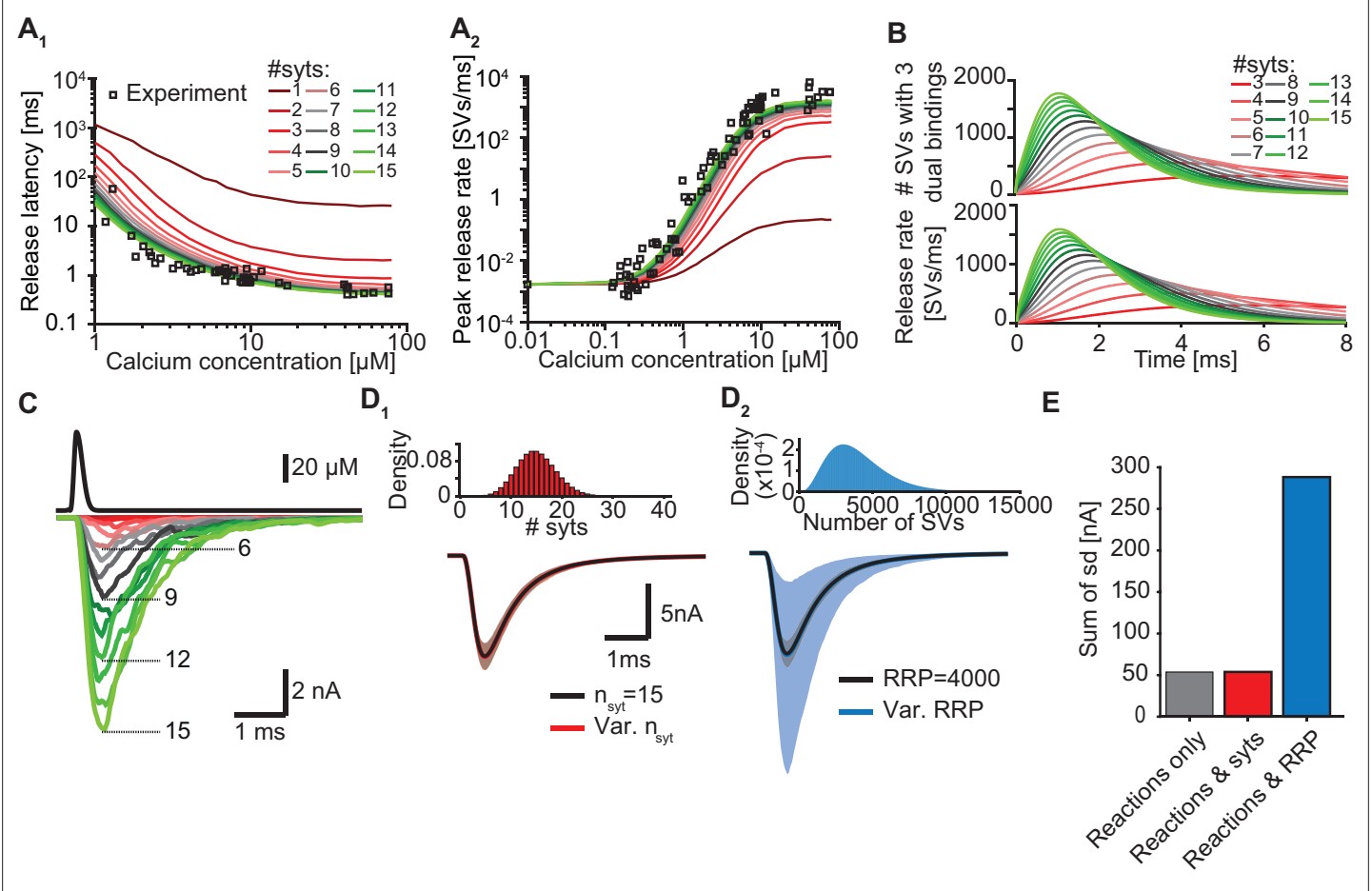

**Figure 5.** Simulations with reduced syt expression predict a reduction in SV fusion. (**A**) Model predictions of median release latencies (**A₁**) and mean peak release rates (**A₂**) as a function of $[Ca^{2+}]_i$ for different numbers of syts per SV. All simulations were performed with 1000 repetitions using the best fit parameters obtained by fitting with $n_{syts} = 15$ and $M_{slots} = 3$. Experimental data points are replotted from *Kochubey and Schneggenburger, 2011*. (**B**) The average number of SVs with three dual bindings formed (top) and release rate (bottom) as a function of time for 3–15 syt copies per SV from simulations with a Ca²⁺ flash of 50 µM. (**C**) Predicted AP-evoked responses (bottom) simulated using a realistic Ca²⁺ transient (top) (*Wang et al., 2008*) for different numbers of syts per SV. The AP-evoked response shown at the bottom are representative single stochastic simulations with an amplitude closest to the mean amplitude of 200 repetitions. Increasing $[PI(4,5)P_2]$ for each choice of $n_{syt} \geq 3$ could fully rescue release latencies, release rates, and evoked responses (*Figure 5—figure supplement 2*). (**D**) Variability in simulated AP-evoked responses for a model with a variable number of syts and an RRP size of 4000 (**D₁**, red, bottom) and a model with a variable RRP size and 15 syts per SV (**D₂**, blue, bottom) compared to the variance induced by the stochasticity of the reactions only (with fixed number of SVs and syts, grey). Solid lines depict mean traces and the shaded area indicates the 95% prediction interval. Simulations were with 1000 repetitions. Top panels show the probability density distributions of the number of syts (Poisson distribution, lambda = 15) and of the number of SVs (gamma distribution, mean of 4000 SVs, standard deviation of 2000 SVs, outcome rounded to nearest integer). (**E**) Quantification of the variance in the traces introduced by the stochasticity of the model reactions (grey), by the stochasticity of model reactions and variable syt number (red), and by the stochasticity of model reactions and variable RRP size (blue) by computing the sum of the standard deviation (sd) determined over the entire trace (0–6ms, 300 data points). Simulation of the individual syts using the Gillespie algorithm agreed with simulations using the analytical solution of the Markov model (*Figure 5—figure supplement 1* and Methods). Simulation scripts can be found in *Source code 1*. Simulation results shown in this figure can be found in *Figure 5—source data 1*.

The online version of this article includes the following source data and figure supplement(s) for figure 5:

**Source data 1.** Source data for *Figure 5* and *Figure 5—figure supplement 1*, *Figure 5—figure supplement 2*.

**Figure supplement 1.** Comparing the two different model implementations.

**Figure supplement 2.** Upregulation of PI(4,5)P₂ can compensate for loss of syts.

wave that RRP SVs experience (*Figure 5C*, top panel)(*Wang et al., 2008*). With 15 syts per SV, AP-induced EPSCs were large and synchronous, but reducing their number decreased response amplitudes (*Figure 5C*). Removal of one syt already reduced the average EPSC amplitude by ~10% and removal of half (7/15) of its copies reduced it by ~72% (*Figure 5—figure supplement 2*, for representative

example traces see *Figure 5C*). Note, however, that our model only describes the functioning of syt1 /syt2 and therefore does not include other $Ca^{2+}$ sensors, like syt7 and Doc2B, which may mediate release in case of syt1 /syt2 loss (*Bacaj et al., 2013*; *Kochubey et al., 2016*; *Kochubey and Schneggenburger, 2011*; *Luo and Südhof, 2017*; *Sun et al., 2007*; *Wen et al., 2010*; *Yao et al., 2011*).

As the number of syts per SV has a large impact on fusion kinetics, we wondered to what extent fluctuations in the number of syts per SV affected the variance in AP-evoked responses in case of their imperfect sorting. Strikingly, however, varying the number of syts per SV over a large range (Poisson distribution with mean = 15, *Figure 5D1, E*) did not increase the variability of AP-evoked in synaptic responses while fluctuations of the RRP size strongly impacted them (*Figure 5D2, E*). This shows that although release kinetics strongly depend on the average number of syts per SV, the system is rather insensitive to fluctuations around this number between individual SVs. Taken together, our data show that although only a subset of syts are required to simultaneously bind $Ca^{2+}$ and $PI(4,5)P_2$ to induce fusion, all SV syts contribute to the high rates of NT release by increasing the probability that several syts simultaneously engage in dual $Ca^{2+}/PI(4,5)P_2$ binding.

Besides the number of syts, the $PI(4,5)P_2$ levels also determine how likely it is for syts to engage in dual $Ca^{2+}/PI(4,5)P_2$ binding at an SV (see *Figure 3* and *Figure 2—figure supplement 7*). We therefore reasoned that upregulation of $PI(4,5)P_2$ levels, which are dynamically regulated (*Jensen et al., 2022*), could potentially compensate for reduced syt expression. To investigate this, we refitted the models with reduced syt levels to the experimental $Ca^{2+}$ uncaging data (*Kochubey and Schneggenburger, 2011*) and only allowed the $PI(4,5)P_2$ concentration in the slots ($[PI(4,5)P_2]$) to vary. Strikingly, increasing $[PI(4,5)P_2]$ fully rescued the characteristics of NT release upon reductions in syt levels down to 3 syts per SV (corresponding to an 80% reduction) by restoring the number and speed of C2B domains engaging in $Ca^{2+}/PI(4,5)P_2$ dual binding (*Figure 5—figure supplement 2A–C*). The required increase in $[PI(4,5)P_2]$ ranged froma factor ~1.1 (14 syts) toa factor ~10 (3 syts, *Figure 5—figure supplement 2D*). These elevations also fully restored simulated AP-evoked responses when at least three syts per SV were present (*Figure 5—figure supplement 2E, F*). Altogether, these data indicate that upregulating $[PI(4,5)P_2]$ is a potential, powerful compensatory mechanism to rescue reductions of NT release in case the number of (functional) syts per SV is reduced to no less than three. We note that this compensatory mechanism may strongly influence experimentally observed effects of stoichiometric changes.

## Evaluation of mutants affecting $Ca^{2+}$ binding to the C2B domain reveals diverse effects on AP-evoked transmission

$Ca^{2+}$ sensing of syts depends on negatively charged aspartate (D) sidechains of the C2B domain whose positions are optimal to bind $Ca^{2+}$ ions (*Fernandez et al., 2001*). The local negative charges of the $Ca^{2+}$ binding sites are reduced/neutralized upon $Ca^{2+}$ binding. The $Ca^{2+}$ binding pockets of the C2B domain have been extensively studied using various mutations (*Bradberry et al., 2020*; *Guan et al., 2017*; *Kochubey and Schneggenburger, 2011*; *Lee et al., 2013*; *Mackler et al., 2002*; *Nishiki and Augustine, 2004*; *Shin et al., 2009*). Mutations that remove or invert the negative charge of the $Ca^{2+}$ binding sites (by mutation to asparagine (N) or lysine (K), 'DN' or 'DK') block $Ca^{2+}$ binding and severely reduce exocytosis, even when co-expressed together with the wildtype protein (*Bradberry et al., 2020*; *Kochubey and Schneggenburger, 2011*; *Lee et al., 2013*; *Mackler et al., 2002*). Other mutations also interfere with $Ca^{2+}$ binding and exocytosis but hold the same pocket charge (e.g. mutation to Glutamate, 'DE') (*Bradberry et al., 2020*). While both types of mutations may similarly interfere with $Ca^{2+}$ binding, they may differentially affect the allosteric mechanism. The allosteric coupling between the $Ca^{2+}$ and $PI(4,5)P_2$ binding sites might be (in part) mediated by electrostatic interactions (*van den Bogaart et al., 2012*), which would imply that the negatively charged $Ca^{2+}$ binding pocket repels $PI(4,5)P_2$ until $Ca^{2+}$ reverses the electrostatic charge, and vice versa. Following this assumption, charge-altering mutations within the $Ca^{2+}$ binding pockets ('DN', 'DK') would partially activate the allosteric coupling mechanism and thereby affect the domain's $PI(4,5)P_2$ affinity (which would not be the case in mutants conserving the charges ('DE')). We explored this possibility in our model using two different hypothetical $Ca^{2+}$ site mutants (*Figure 6A*). We investigated the effect of these mutants on AP-induced synaptic transmission under homozygous and heterozygous expression conditions (combined expression of mutant and WT with a total of 15 syts per SV; *Figure 6B*).

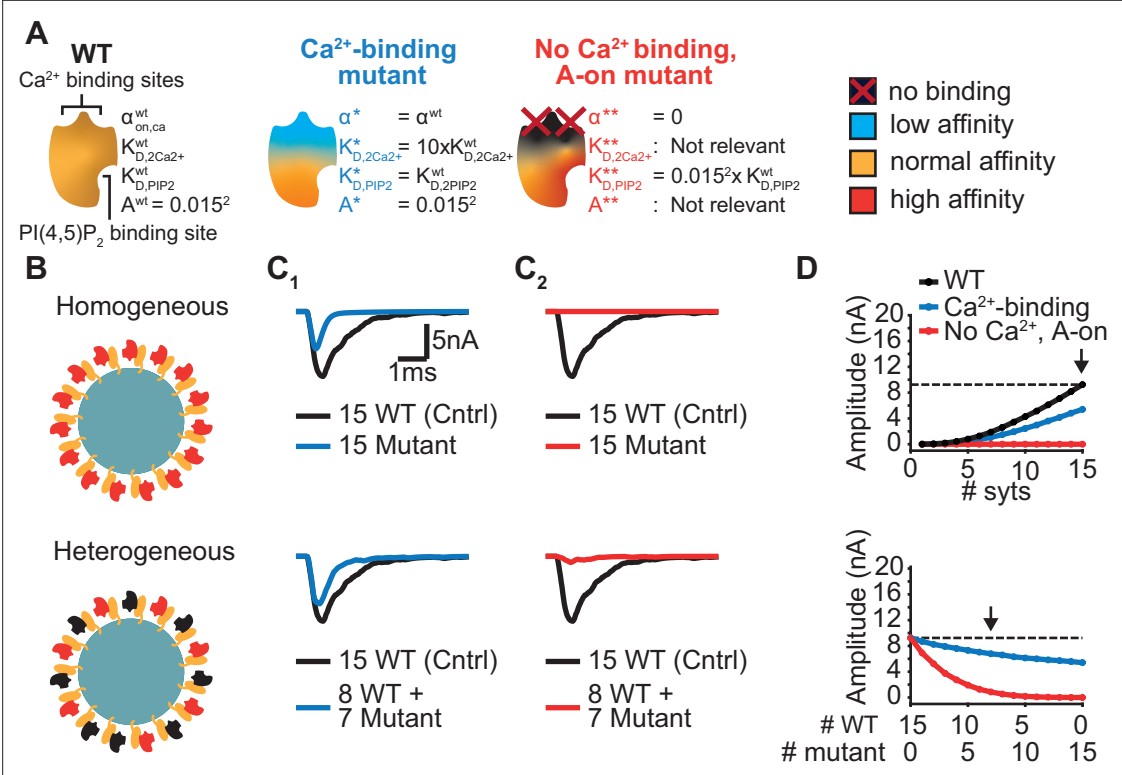

**Figure 6.** Systematic evaluation of the effect of mutant syts on simulated AP-evoked fusion. (**A**) Illustration of a WT syt and two mutant syts. The "*Ca²⁺-binding*" mutant has a lower affinity for Ca²⁺ (K$_{D,2Ca2+}$ 10x increased, that is β 10 x increased). The "*no-Ca2²⁺ binding, A-on*" mutant is not able to bind Ca²⁺ and has a high binding affinity for PI(4,5)P₂, which is equal to the affinity for PI(4,5)P₂ when the allostericity between Ca²⁺ and PI(4,5)P₂ is "active" in WT syts (Ca²⁺-bound state). Because of the inability to bind Ca²⁺, allosteric interactions between Ca²⁺ and PI(4,5)P₂ are not possible in this mutant. (**B**) Illustration of homogeneous (top) and heterogeneous expression (bottom) of the mutants. Mutant syts are depicted in red, WT syts are depicted in black. (**C**) Representative, stochastically simulated AP-evoked responses with homozygous (top, 15 mutant syt copies) and heterozygous (bottom, 8 WT and 7 mutant syt copies) expression of the different mutants (C₁: '*Ca²⁺-binding*' mutant, in blue; C₂: '*no Ca²⁺ binding, A-on*' mutant, in red). For each of the settings a representative trace of a condition with 15 WT syts is shown in black (control condition). A third mutation, the "*no Ca²⁺ binding, A-off*" was also explored (*Figure 6—figure supplement 1*). (**D**) Mean amplitudes of simulated AP-evoked responses (n=200) for the homogeneous (top) and heterogeneous (bottom) expression of the different mutants, and for WT syt (for homozygous condition only). Dotted line indicates the mean amplitude of simulated eEPSCs with 15 copies of WT syt (control). Arrow indicates the condition that is depicted in panels C. Simulation scripts can be found in *Source code 1*. Results from simulations can be found in *Figure 6—source data 1*.

The online version of this article includes the following source data and figure supplement(s) for figure 6:

**Source data 1.** Source data for *Figure 6* and *Figure 6—figure supplement 1*.

**Figure supplement 1.** The dominant negative effect of a mutant that is unable to bind Ca2²⁺ depend on the mutants PI(4,5)P₂ affinity.

The first mutant, the '*Ca²⁺-binding*' mutant, had a lower Ca²⁺ affinity (10xK$_{D,2Ca2+}$), but all other properties were the same as in the WT C2B domain. This might be similar to a mutant with a mutation of the binding pocket which conserves its charge (e.g. 'DE'). When homozygously expressed, this mutant showed eEPSCs with a~50% reduced amplitude and faster kinetics compared to the WT condition (*Figure 6C1*, *Figure 6D* top). Heterozygous expression, with 8 WT and 7 mutant syts per SV, only caused a small decrease in mean eEPSC amplitude compared to the expression of 15 WT syts per SV (*Figure 6C1–D*, bottom), showing that this mutant is relatively mild.

The second hypothetical mutation was designed to not only abolish Ca²⁺ binding, but to also mimic the Ca²⁺-bound state. Thereby, this mutant featured a high PI(4,5)P₂ affinity as if the allosteric interaction between Ca²⁺ and PI(4,5)P₂ was permanently 'on'. This might represent an extreme example of a mutation electrostatically reducing/inverting the negative charges of the Ca²⁺ binding pocket (e.g. 'DN', 'DK'). We termed this mutant the '*no Ca²⁺ binding, A-on*' mutant (*Figure 6A*). This mutant showed no NT release in response to the Ca²⁺ transient in a homozygous condition (*Figure 6C2–D*, top), which is explained by its inability to bind Ca²⁺. A major detrimental effect of the mutant was

observed when co-expressed with the wildtype protein: When half of the syts on the SV were mutated (heterozygote), the amplitude of simulated eEPSCs was strongly reduced (*Figure 6C2–D*, bottom). Merely four mutant proteins expressed together with 11 WT proteins already decreased eEPSC amplitudes by ~70% (*Figure 6D*, bottom), indicating a strong dominant negative effect. The strong inhibition is a result of the mutant's increased PI(4,5)P$_2$ affinity leading to occupation of PI(4,5)P$_2$ binding slots on the membrane with this Ca$^{2+}$-insensitive mutant which blocks the association of the Ca$^{2+}$ sensitive- and SV fusion-promoting WT proteins. In comparison, a mutant not able to bind Ca$^{2+}$ but having a normal PI(4,5)P$_2$ affinity ('*no Ca$^{2+}$ binding, A-off*' mutant, which could represent a more extreme form of the 'DE' mutant) had a much weaker effect (*Figure 6—figure supplement 1*). This indicates that the allosteric interaction between Ca$^{2+}$ and PI(4,5)P$_2$ can play a prominent role in the severity of mutations.

## Rapid changes of accessible PI(4,5)P$_2$ dramatically impact synaptic short-term plasticity

In our model, we describe the PI(4,5)P$_2$ levels in concentration units, because our model is based on syt affinities for Ca$^{2+}$ and PI(4,5)P$_2$ determined in vitro (*van den Bogaart et al., 2012*). The estimated concentration of PI(4,5)P$_2$ not only depends on the local density of PI(4,5)P$_2$ in the membrane, but also on the accessibility syt has to PI(4,5)P$_2$. While all species (Ca$^{2+}$, PI(4,5)P$_2$, and syt C2B) are homogenously accessible in the aqueous solution of the in vitro setting (*van den Bogaart et al., 2012*), at the synapse the syt C2 domains have constrained motility due to their vesicular association and PI(4,5)P$_2$ is restricted to (clusters on) the plasma membrane (*Milosevic et al., 2005; van den Bogaart*

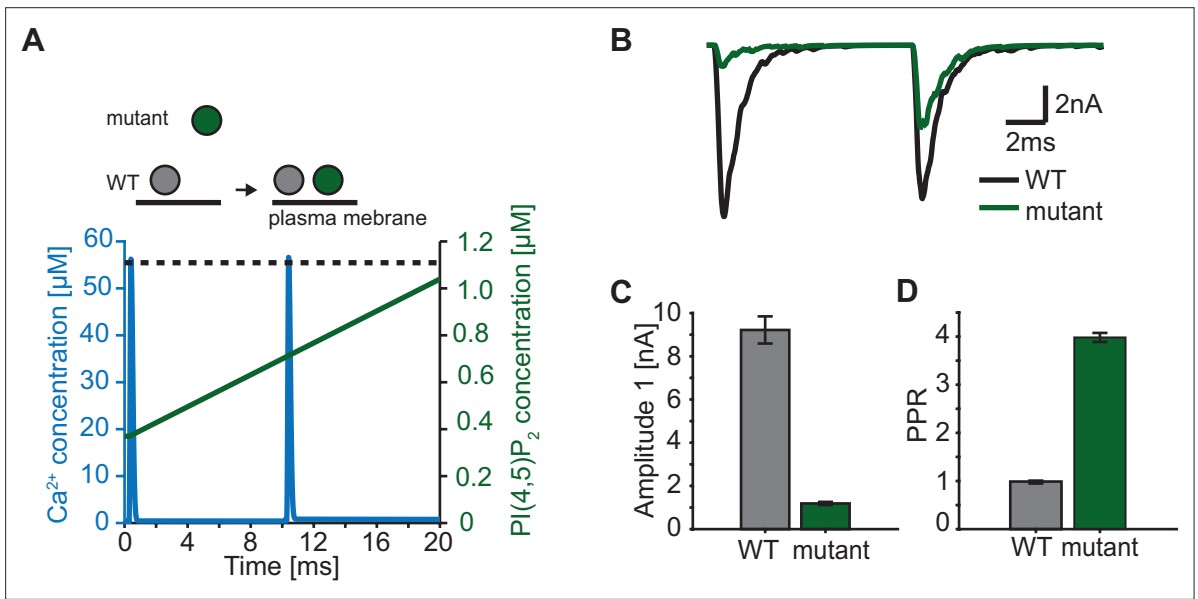

**Figure 7.** Paired-pulse stimulation in a membrane binding syt mutant. (**A**) Time course of [Ca$^{2+}$]$_i$ (blue) and [PI(4,5)P$_2$](dashed black line: wildtype (WT), green line: mutant). Top panel illustrates the placement of vesicles with respect to the PM for SVs expressing WT syt (grey SVs) and SVs expressing a syt mutant deficient in membrane binding (green SVs, homozygous expression) before the first (left side of arrow) and second AP (right side of arrow). In WT conditions, most SVs reside close to the PM before the onset of the first stimulus. Before onset of the second pulse, WT SVs keep the same average distance to the PM. Mutant SVs, however, show a large distance to the PM at the onset of the first stimulus. Before the onset of the second AP, mutant SVs move closer to the PM due to a Ca$^{2+}$-dependent mechanism (*Chang et al., 2018*). The bottom panel shows the Ca$^{2+}$ (blue) and PI(4,5)P$_2$ (green) transients over time in a paired pulse stimulus (10ms between stimuli). Due to the increased distance between the SV and the PM in the membrane binding mutant, mutant SVs are assumed to experience a lower [PI(4,5)P$_2$] (solid green line) compared to WT SVs (dotted, black line) at the start of the first stimulus. Before the start of the second stimulus, mutant SVs move closer to the PM which increases the experienced [PI(4,5)P$_2$] of these SVs. (**B**) Representative eEPSCs simulated using the Ca$^{2+}$ and PI(4,5)P$_2$ transients depicted in A. (**C**) Amplitude of the first eEPSC for WT and mutant. (**D**) Paired-pulse ratio (PPR) for WT and mutant. Data in C and D show mean ± SEM, using 50 repetitions and a variable RRP size (see Materials and methods for details, the same RRP values were used for both the mutant and the WT condition). Simulation scripts can be found in *Source code 1*. Depicted simulation results can be found in *Figure 7—source data 1*.

The online version of this article includes the following source data for figure 7:

**Source data 1.** Source data for *Figure 7*.

*et al., 2011a*). This implies that the positioning of SVs with respect to the plasma membrane has an impact on the PI(4,5)P$_2$ concentration accessible to syts. We so far assumed that all syts of RRP SVs are exposed to the same PI(4,5)P$_2$ levels. This could be the case if all SVs are similarly docked to the plasma membrane. However, when considering more complex stimulation paradigms such as repetitive AP stimulation, this may no longer be valid as several studies reported activity-dependent changes in SV positioning on a millisecond timescale (*Chang et al., 2018*; *Kusick et al., 2020*; *Miki et al., 2016*). Rapid changes in accessible PI(4,5)P$_2$ may thus contribute to short-term plasticity, the alteration of responses on a millisecond timescale (*Abbott and Regehr, 2004*; *Kobbersmed et al., 2020*; *Neher and Brose, 2018*; *Silva et al., 2021*). Recent studies reported that mutations of positively charged amino acids of the C2B domain (lysines, 'Ks', implicated in binding of PI(4,5)P$_2$ and/or the SNAREs and arginines, 'Rs', implicated in binding the plasma membrane and/or the SNAREs) resulted in a loss of SV docking and severely reduced neurotransmission (*Chang et al., 2018*; *Chen et al., 2021*; *Li et al., 2006*; *Xue et al., 2008*). Strikingly, SV docking in these mutants was rapidly restored within milliseconds after an AP which also led to enhanced synaptic transmission in response to a second AP given 10ms after the first (*Chang et al., 2018*). We explored such a situation in the context of our model by driving exocytosis with two successive AP-induced Ca$^{2+}$ transients and assuming either constant PI(4,5)P$_2$ levels for syts in wildtype synapses (i.e. all RRP SVs similarly docked) or initially reduced and activity-dependent increasing PI(4,5)P$_2$ levels for syts in mutant synapses (where SVs docked after the first AP; *Figure 7A*). We studied the consequence of a mutation that would only affect SV docking at steady state (as may be the case upon mutation of the arginines 398 and 399 of mouse syt 1, 'R398,399Q') in (*Figure 7*). This resulted in a markedly decreased initial response (*Figure 7B and C*), but repeated activation induced a large facilitation of responses (indicated by a large paired pulse ratio: quotient of the second EPSC amplitudes divided by the first) (*Figure 7D*). We conclude that dynamic changes in the PI(4,5)P$_2$ levels accessible to syts – which may be caused by activity-dependent SV relocation – strongly impact synaptic short-term plasticity. Mutations of the C2B domain that reduce its PI(4,5)P$_2$ affinity (as is likely the case upon mutation of the lysine residues 325 and 327 in syt1 or 327, 328 and 332 in syt2) may be more detrimental because even when the effective PI(4,5)P$_2$ concentration accessible to syts is restored upon activity-dependent SV redocking, syt association to PI(4,5)P$_2$ will still be less probable.

## Discussion

Here, we propose a quantitative, experiment-based model describing the function of syt in SV fusion on a molecular level based on biochemical properties determined in vitro. In our model, syt acts by lowering the energy barrier for SV fusion by dual binding to Ca$^{2+}$ and PI(4,5)P$_2$. When allowing at least three dual-bound syts per SV at a time, this model can explain the steep Ca$^{2+}$ dependence of NT release observed at the calyx of Held synapse (*Kochubey and Schneggenburger, 2011*). Exploring this model led to the following conclusions:

1. The positive allosteric interaction between Ca$^{2+}$ and PI(4,5)P$_2$ is crucial for fast SV fusion as it stabilizes the dual-bound state which allows multiple syts to successively lower the energy barrier for SV fusion;
2. At least three slots per SV for syt Ca$^{2+}$/PI(4,5)P$_2$ dual binding are needed to achieve the speed and Ca$^{2+}$ sensitivity inherent to synaptic transmission;
3. Only few syts (≤4) work together for fast SV fusion on most SVs.
4. A high copy number of syts per SV ensures fast NT release by increasing the probability that several syts engage in Ca$^{2+}$/PI(4,5)P$_2$ dual binding;
5. Binding of syts to PI(4,5)P$_2$ prior to the Ca$^{2+}$ stimulus allows some SVs to fuse very fast (submillisecond).
6. The molecular resolution of this model can be used to study consequences of mutations.

### A syt-dependent switch on the energy barrier for SV fusion

Exocytosis is a highly energy-demanding reaction, for which the formation of the neuronal SNARE complex provides the energy (*Jahn and Fasshauer, 2012*). In our model we assume that syts regulate this process by lowering the activation energy barrier for exocytosis when they engage in Ca$^{2+}$/PI(4,5)P$_2$ dual binding. However, how Ca$^{2+}$/PI(4,5)P$_2$ dual binding to syt exactly reduces this energy barrier

is not known. One possibility is that the energy is provided by the SNAREs themselves and that $Ca^{2+}/PI(4,5)P_2$ dual binding to syt relieves a clamping function, which syt itself or the auxiliary protein complexin exerts on the SNAREs (*Courtney et al., 2019*; *Schupp et al., 2017*; *Tang et al., 2006*; *Zhou et al., 2015*; *Zhou et al., 2017*). Alternatively – or additionally – syt's dual binding $Ca^{2+}/PI(4,5)P_2$ might promote SNARE-mediated fusion by changing the local electrostatic environment (*Ruiter et al., 2019*; *Shao et al., 1997*). Furthermore, dual-binding syts could bring SVs closer to the plasma membrane, potentially below an upper limit for full SNARE complex assembly (*Araç et al., 2006*; *Chang et al., 2018*; *Honigmann et al., 2013*; *Hui et al., 2011*; *Lin et al., 2014*; *Nyenhuis et al., 2019*; *van den Bogaart et al., 2011b*; *Xue et al., 2008*). In line with these hypothetical working mechanisms, our estimated effect each dual-bound C2B domain has on the energy barrier height (~5 $k_bT$) is similar to the estimated energy barrier height for the final zippering step of the SNARE complex (*Li et al., 2016*). Syt's $Ca^{2+}/PI(4,5)P_2$ dual binding might also promote fusion by inducing membrane curvature or favoring lipid rearrangement (*Lai et al., 2011*; *Martens et al., 2007*). In line with this reasoning, our estimated contribution of a syt engaging in $Ca^{2+}/PI(4,5)P_2$ dual binding is similar to estimates of syt1 membrane binding energies (*Gruget et al., 2020*; *Gruget et al., 2018*; *Ma et al., 2017*). In our model, we assume that multiple syts can simultaneously reduce the energy barrier for fusion. Here we assumed that all syts exert the same effect on this energy barrier and that the effects of more dual-bound syts are additive. Whether or not this is the case will depend on the precise mechanism by which they shape the energy landscape. We show here that the simplest model (constant and independent contribution) is sufficient to reproduce the biological response.

Both the C2A and C2B domain of syt cooperate in SV exocytosis (*Bowers et al., 2020*; *Gruget et al., 2020*; *Lee et al., 2013*; *Wu et al., 2021b*). However, the exact role of the C2A domain in triggering SV fusion remains debated (*Fernández-Chacón et al., 2002*; *Lee et al., 2013*; *Paddock et al., 2011*; *Sørensen et al., 2003*; *Stevens and Sullivan, 2003*; *Striegel et al., 2012*). As mutation of the $Ca^{2+}$ binding pockets of the C2A domain did not affect the affinities of $Ca^{2+}$ and $PI(4,5)P_2$ in vitro (*van den Bogaart et al., 2012*), we focused on the C2B domain in our model. Moreover, we aimed at developing a minimal molecular model with the least number of parameters that can fully recapitulate physical responses of the synapse. This, however, does not exclude the possibility that our C2B domain-based model indirectly describes properties of the C2A domain. For instance, $Ca^{2+}$ binding to the C2A domain may influence the $Ca^{2+}$ affinity of the C2B domain (*Sørensen et al., 2003*). This property may affect the values of other parameters of the model (e.g. our estimate of the $PI(4,5)P_2$ concentration), meaning that these effects might be captured indirectly when fitting experimental data.

## Allostericity buys time to synchronize syts

Our modeling study proposes that the allostericity between $Ca^{2+}$ and $PI(4,5)P_2$ binding is essential for the syts to achieve fast, synchronous, and sensitive NT release (*van den Bogaart et al., 2012*, *Figure 4—figure supplement 1*). With their experiment, *van den Bogaart et al., 2012* determined steady state affinities, which do not provide information on the association/dissociation rates. This means that the allosteric effect may either be due to speeding up the association or slowing down the dissociation of $Ca^{2+}/PI(4,5)P_2$ (*Figure 1A*). Here we implemented the latter, a reduction of the unbinding rates of both $Ca^{2+}$ and $PI(4,5)P_2$ when both species were bound to the C2B domain, which leads to a stabilization of the dual-bound state. A stabilization of the $Ca^{2+}$-bound states was also essential to reproduce the $Ca^{2+}$ dependence of release in the previously proposed five-site binding model (*Heidelberger et al., 1994*; *Schneggenburger and Neher, 2000*). Here we show in the context of our model that increasing the lifetime of $Ca^{2+}/PI(4,5)P_2$ dual binding is particularly important to achieve fast fusion rates as it allows several C2B domains to simultaneously engage to lower the fusion barrier (*Figure 4*). The drawback of the strong allosteric interaction between the $Ca^{2+}$ and $PI(4,5)P_2$ bindings sites might be its potential involvement in the strong dominant-negative effects of some C2B domain mutations (*Figure 6*).

## The stoichiometry of the SV fusion machinery

Each SV contains multiple syt copies (*Takamori et al., 2006*), which can jointly participate in the fusion process. However, the number of syts that can simultaneously engage with $PI(4,5)P_2$ located at the plasma membrane, and thus can cooperate during fusion, is likely limited. There are several possible

explanations for this limit. First, the space between the vesicular and plasma membrane is limited and crowded by many synaptic proteins (*Wilhelm et al., 2014*). In addition, plasma membrane association of syt may require interaction with the SNAREs (*de Wit et al., 2009*; *Mohrmann et al., 2013*; *Rickman and Davletov, 2003*; *Zhou et al., 2015*), which limits the number of association points. Moreover, the inhomogeneous distribution of PI(4,5)P$_2$ in the plasma membrane might put further constraints on association of syt to PI(4,5)P$_2$ (*Milosevic et al., 2005*; *van den Bogaart et al., 2011a*). Other proteins able to promote SV fusion, like Doc2, might also rely on this limited number of membrane contact points/resource and compete with syt. Our model predicts that most SVs already bind one or two slots with syt at rest (*Figure 3*), and this might explain the ability of syt to clamp spontaneous transmission (*Bouazza-Arostegui et al., 2022*; *Courtney et al., 2019*; *Kochubey and Schneggenburger, 2011*; *Schupp et al., 2017*).

We found that at least three PI(4,5)P$_2$ association sites ('slots') were required to explain the steep Ca$^{2+}$ dependency of neurotransmitter release (*Figure 2A–C*). These findings are compatible with a cryo-EM analysis that identified six protein complexes between docked SVs and plasma membrane (*Radhakrishnan et al., 2021*). Interestingly, irrespective of the number of slots for models with three or more slots, our analysis suggests that most fusion events at [Ca$^{2+}$]$_i$ > 1 µM occurred after engaging three syts in Ca$^{2+}$/PI(4,5)P$_2$ dual binding (*Figure 2D–E*). At lower [Ca$^{2+}$]$_i$ (0.5–1 µM, *Figure 2—figure supplement 2B* and *Figure 2—figure supplement 3B*), the number of dual bindings leading to fusion was reduced to 1–2, indicating that higher [Ca$^{2+}$]$_i$ recruits additional syts to increase fusion rates. Although our model indicates that only few syts are involved in fusion, more syts could be involved in upstream reactions.

The predicted number of three syts involved in fast exocytosis matches experimental estimates of the number of SNARE-complexes zippering for fast vesicle fusion (*Arancillo et al., 2013*; *Mohrmann et al., 2010*; *Shi et al., 2012*; *Sinha et al., 2011*; but higher estimates in the number of SNARE complexes actively involved in fusion have also been reported *Wu et al., 2017*). Moreover, our model is consistent with a previous model of neurotransmitter release at the frog neuromuscular junctions that estimated that fusion is triggered by the binding of two Ca$^{2+}$ ions to each of three syts (*Dittrich et al., 2013*). That model, which describes Ca$^{2+}$ dynamics in the AZ in detail, showed that many additional Ca$^{2+}$ binding sites (20-40) were required to enhance fusion probability, because the probability of having a single Ca$^{2+}$ molecule in the vicinity of SVs is extremely low. Similarly, our model predicts a relevance of a high vesicular syt copy number, because, even though fusion involves only a handful of syts, many copies per SV are necessary to speed up the collision with multiple slots (*Figure 5*). In fact, high protein abundance could play a general role in promoting collision-limited processes in SV fusion, and may provide an intuitive explanation for the many (~70) synaptobrevins on SVs which may assemble into SNARE complexes downstream of syt action (*Takamori et al., 2006*; *van den Bogaart et al., 2011b*).

Our model of dynamic assembly of multiple C2B domains in Ca$^{2+}$/PI(4,5)P$_2$ dual binding in response to Ca$^{2+}$ is fundamentally different from studies suggesting that 12–20 syts need to preassemble in higher-order complexes (rings) to execute their function in fusion (*Rothman et al., 2017*). A testable property to distinguish these possibilities is the sensitivity to reducing the number of syts per SV. If SV fusion relied on preassembled syt-rings, it would immediately break down if the number of syts was reduced to a number preventing ring assembly, whereas our model predicts gradual effects of reduced syt copy numbers (even for titration below $n_{syt}$ = 3; *Figure 5*). In line with the latter case, recent experiments that reported that SV fusion is rather sensitive to progressive reductions in syt levels (*Bouazza-Arostegui et al., 2022*). On the other hand, it might be sufficient for syts to occupy fewer slots at rest (1–2, *Figure 3*) to exert their effects on SV priming and clamping of spontaneous release (which are not included in our model), which might explain why these reactions appear less sensitive to syt1 reductions (*Bouazza-Arostegui et al., 2022*).

## Heterogeneity in PI(4,5)P$_2$ concentration between different RRP SVs

The interaction between syt and PI(4,5)P$_2$ has been shown to be essential in SV exocytosis (*Bai et al., 2004*; *Li et al., 2006*; *Wu et al., 2021a*), but also has been found to play a role in SV docking (*Chang et al., 2018*; *Chen et al., 2021*). Consistently, we observed that at resting synapses the majority of SVs (~83%) contain at least one syt bound to PI(4,5)P$_2$ in our model (*Figure 3*). The number of syts bound to PI(4,5)P$_2$ per SV at rest highly influenced the release probability, leading to heterogeneity within the

RRP (*Figure 3*, *Wölfel et al., 2007*). As PI(4,5)P$_2$ levels have a large impact on release kinetics (shown in this study, but also by *Walter et al., 2017*), heterogeneity between RRP SVs might further be enhanced by unequal PI(4,5)P$_2$ levels. Additionally, the strong impact of PI(4,5)P$_2$ levels on SV fusion indicates that the dynamic regulation of PI(4,5)P$_2$ occurring at the seconds time scale might strongly influence synaptic plasticity (*Jensen et al., 2022*).

In our model, we described PI(4,5)P$_2$ levels in concentration units to constrain our model by using in vitro PI(4,5)P$_2$ affinity measurements (*van den Bogaart et al., 2012*). However, this concentration does not only encompass the density of PI(4,5)P$_2$ in the plasma membrane, but also includes the accessibility of syt to PI(4,5)P$_2$. Several studies have shown that PI(4,5)P$_2$ is distributed heterogeneously over the plasma membrane in clusters that contain a high PI(4,5)P$_2$ density (*Honigmann et al., 2013*; *Milosevic et al., 2005*; *van den Bogaart et al., 2011a*). Moreover, syts located closer to the plasma membrane will have increased access to PI(4,5)P$_2$ compared to those located further away. Taken together, this indicates that the PI(4,5)P$_2$ concentration is likely to vary between RRP SVs and also between individual syts on the SV. Furthermore, this implies that once a syt has engaged in PI(4,5)P$_2$ binding the successive engagement of additional syts might be favored for some (those facing towards the PM) and disfavored for others (those facing from the PM). While knowledge of these details could be helpful to construct a more realistic version of our molecular model, we currently do not possess the methodology to measure these properties. Therefore, in our model, we simulated the simplest scenario where all syts have an equal probability of engaging in PI(4,5)P$_2$ binding.

As the localization of syts with respect to the PM influences the accessibility of syt to PI(4,5)P$_2$, mutations in synaptic proteins and stimulation protocols that alter SV docking will affect the PI(4,5)P$_2$ concentration as it is implemented in our model (*Chang et al., 2018*; *Chen et al., 2021*; *Kusick et al., 2020*). Using a time-dependent PI(4,5)P$_2$ concentration, we illustrated the impact this might have on the short term plasticity of synaptic responses (*Figure 7*). This is a simplification, as we did not take the individual SV/syt distances to the PM into account. This distance is affected by several synaptic proteins, including syt1, Munc13, and synaptotagmin7 (*Chen et al., 2021*; *Imig et al., 2014*; *Liu et al., 2016*; *Quade et al., 2019*; *Tawfik et al., 2021*; *Voleti et al., 2017*). A role of these proteins in short-term plasticity is firmly established, yet precise mechanistic details are still lacking (*Jackman et al., 2016*; *Rosenmund et al., 2002*; *Shin et al., 2010*). The extension of models based on molecular interactions such as presented here should allow reproduction of responses to more complex synaptic activity patterns relevant for neural processing. Particularly the molecular resolution of such models will be useful to conceptualize the importance of specific molecular interactions for physiological and pathological processes at the synapse.

## Materials and methods

In this paper, we propose a model for SV fusion induced by Ca$^{2+}$ and PI(4,5)P$_2$ binding to $n_{syts}$ syts per SV, with at most $M_{slots}$ syts per SV engaging in PI(4,5)P$_2$ binding at the same time. We implemented the model in two ways for different simulation purposes: (1) an implementation based on the analytical solution of the model, and (2) an implementation following the Gillespie algorithm (*Gillespie, 2007*) (Matlab procedures for simulations can be found in *Source code 1*). In the first implementation, we assume a constant [Ca$^{2+}$]$_i$ (allowing us to simulate Ca$^{2+}$ uncaging experiments: SV reactions at a Ca$^{2+}$ level reached by the uncaging stimulus from a steady state starting point calculated for the resting Ca$^{2+}$ concentration of 50 nM), whereas the second version was implemented to allow for [Ca$^{2+}$]$_i$ to vary over time (allowing us to simulate AP-evoked responses). Another important difference between the two approaches is that the analytical solution describes the binding state of an entire SV and the Gillespie version describes the binding state of each individual syt. Both implementations allow for stochastic evaluation of the model. The first implementation is used in *Figure 2*, *Figure 2—figure supplement 4*, *Figure 2—figure supplement 5*, *Figure 2—figure supplement 6*, *Figure 2—figure supplement 7*, *Figure 3*, *Figure 3—figure supplement 1*, *Figure 4*, *Figure 4—figure supplement 1*, *Figure 5*, *Figure 5—figure supplement 1*, *Figure 5—figure supplement 2*. The second implementation is used to simulate the AP-evoked responses and individual SV binding states in *Figure 2—figure supplement 2*, *Figure 2—figure supplement 3*, *Figure 2—figure supplement 4*, *Figure 2—figure supplement 5*, *Figure 2—figure supplement 6 Figure 4*, *Figure 5*, *Figure 5—figure supplement 1*, *Figure 5—figure supplement 2 Figure 6*, and *Figure 6—figure supplement 1*, and *Figure 7*.

**Table 3.** Overview of possible reactions and their rates in the model.

| Reaction | Condition | Triplet notation | Reaction rate |
|---|---|---|---|
| Binding of PI(4,5)P$_2$ to unbound syt | $n+m+k<n_{syts}$ and $n+k < M_{slots}$ | $(n,m,k)\to(n,m,k+1)$ | $(n_{syts}-n-m-k)(M_{slots}-n-k)\,[PI(4,5)P_2]\gamma$ |
| Unbinding of PI(4,5)P$_2$ | $k>0$ | $(n,m,k)\to(n,m,k-1)$ | $k\delta$ |
| Binding of Ca$^{2+}_2$ to unbound syt | $n+m+k<n_{syts}$ | $(n,m,k)\to(n,m+1,k)$ | $(n_{syts}-n-m-k)[Ca^{2+}]^2\alpha$ |
| Unbinding of Ca$^{2+}_2$ | $m>0$ | $(n,m,k)\to(n,m-1,k)$ | $m\beta$ |
| Binding of PI(4,5)P$_2$ to form dual binding | $n+k<M_{slots}$ and $m>0$ | $(n,m,k)\to(n+1,m-1,k)$ | $m(M_{slots}-n-k)[PI(4,5)P_2]\,\gamma$ |
| Unbinding of PI(4,5)P$_2$ from a dual binding | $n>0$ | $(n,m,k)\to(n-1,\,m+1,k)$ | $An\delta$ |
| Binding of Ca$^{2+}_2$ to form a dual binding | $k>0$ | $(n,m,k)\to(n+1,m,k-1)$ | $k[Ca^{2+}]^2\alpha$ |
| Unbinding of Ca$^{2+}_2$ from a dual binding | $n>0$ | $(n,m,k)\to(n-1,m,k+1)$ | $An\beta$ |
| Fusion | | $(n,m,k)\to(F)$ | $L_+\cdot f^n$ |

Consistency between the two approaches was validated by comparison of simulation result distributions in quantile-quantile (Q-Q) plots (*Figure 5—figure supplement 1*).

## SV states and possible reactions in the analytical version of the model

In the analytical solution of the model, we describe for each SV the number of syts having bound two Ca$^{2+}$ ions, PI(4,5)P$_2$, or both species. Since syts were assumed to work independently, their order is not relevant, and we therefore do not need to describe the binding state of each individual syt. The possible binding states of an SV are described in *Figure 1—figure supplement 2*. Each state is represented by the triplet $(n,m,k)$, with $k$ denoting the number syts having bound PI(4,5)P$_2$, $m$ denoting the number of syts having bound two Ca$^{2+}$ ions, and $n$ denoting the number of syts having bound both species and thereby having formed a dual binding. $n_{syts}$ is the total number of syts per SV. $M_{slots}$ restricts the number of syts having bound PI(4,5)P$_2$ simultaneously, which includes syts having bound PI(4,5)P$_2$ only ($k$) and those having formed a dual binding ($n$). Taken together, this implies that for all states in the model, it holds that

$$k + m + n \leq n_{syts} \text{ and } k + n \leq M_{slots}$$

We numbered the states systematically following a lexicographic ordering, excluding the states that violate the inequalities described above. To illustrate, we write the ordering of all the states $(m,n,k)$ with $n_{syts} = 3$ and $M_{slots} = 2$:

(0,0,0),(0,0,1),(0,0,2),(0,1,0),(0,1,1),(0,1,2),(0,2,0),(0,2,1),(0,3,0),(1,0,0),(1,0,1),(1,1,0),(1,1,1),(1,2,0),(2,0,0),(2,1,0).

Besides these binding states, an additional state, $F$, describes whether the SV has fused. With $n_{syts} = 15$ and $M_{slots} = 3$, a single SV in our model has 140+1 states. From any state, there are at most 9 possible reactions, one being SV fusion and the other 8 being (un)binding of Ca$^{2+}$ or PI(4,5)P$_2$ to/from a syt. The rates for the possible reactions of a single SV in this model are summarized in *Table 3*. In many cases, only a subset of the 8 (un)binding reactions are allowed because of the inequalities above (noted under 'Condition' in the table).

The reaction rates of (un)binding Ca$^{2+}$ or PI(4,5)P$_2$ are calculated as the number of syts available for (un)binding (computed using $n_{syts}$, $n$, $m$, $k$) times the reaction rate constant ($\alpha$, $\beta$, $\gamma$, $\delta$), and, in the case of binding reactions, times the concentration of the ligand ([PI(4,5)P$_2$] or [Ca$^{2+}$]). We assumed binding of two Ca$^{2+}$ ions to a single C2B domain. In our model, this two-step process is simplified to a single reaction step by taking [Ca$^{2+}$]$_i$ to the power of two. This simplification is reasonable, because we assumed that syt could only associate to the vesicular membrane when two Ca$^{2+}$ ions are bound, and binding of one Ca$^{2+}$ ion would not induce an 'intermediate' association state to the membrane, nor would it affect the allosteric interaction. By simplifying the two Ca$^{2+}$ binding/ubinding steps to one, we indirectly assumed high cooperativity between the two Ca$^{2+}$ binding sites. To account for the limit on the number of syts bound to PI(4,5)P$_2$, the number of available, empty slots, $(M_{slots}-n-k)$, was multiplied on the PI(4,5)P$_2$ binding rates. The fusion rate of the SV is computed by $L_+\cdot f^n$ (similar to *Lou et al., 2005*), with $L_+$ denoting the basal fusion rate and $f$ the factor of increase in fusion rate by each

dual binding being formed. $L_+$ was set to 4.23e-4 s$^{-1}$ to match the release rate measured at low [Ca$^{2+}$]$_i$, given an average size of the RRP of 4000 SVs (see below).

The affinities for Ca$^{2+}$ and PI(4,5)P$_2$ binding to syt were set to previously determined dissociation constants ($K_{D,Ca2+} = \beta/\alpha = 221^2 \mu M^2$, $K_{D,PI(4,5)P2} = \delta/\gamma = 20 \mu M$) obtained using in vitro microscale thermophoresis experiments (*van den Bogaart et al., 2012*). For determination of the dissociation constant of Ca$^{2+}$, van den Bogaart and colleagues assumed binding of a single Ca$^{2+}$ ion to the C2AB domain (*van den Bogaart et al., 2012*). The $K_{D,2Ca2+}$ of the reaction describing binding of two Ca$^{2+}$ ions, in our model, can be computed from the experimentally derived dissociation constant by taking it to the power of two. This was corroborated by re-fitting the experimental data with a hill coefficient of 2, which yielded a similar $K_{D,2Ca2+}$ value of ~221$^2$ (data not shown).

The in vitro experiments revealed a change in syt1 Ca$^{2+}$ affinity upon binding PI(4,5)P$_2$, and vice versa (*van den Bogaart et al., 2012*), indicating a positive allosteric relationship between the two species. We assumed this allosteric effect was due to a stabilization of the dual-bound state by lowering of the unbinding rates of Ca$^{2+}$ and PI(4,5)P$_2$ with a factor (A=(3.3/221)$^2$ = 0.00022) and occurs when both species have bound. Upon dual binding, both rate constants for unbinding Ca$^{2+}$ and PI(4,5)P$_2$ are multiplied by A, since any closed chemical system must obey microscopic reversibility (*Colquhoun et al., 2004*). Using the biochemically defined affinities, the number of free parameters in our model was constrained to:

$$\xi = \left(\alpha, \; \gamma, \; \left[PI\left(4,5\right)P_2\right], f\right)$$

The values of $\beta$ and $\delta$ were determined according to the affinities for each choice of $\alpha$ and $\gamma$.

## The steady state of the system

The steady state of the system before stimulation was determined at a resting, global [Ca$^{2+}$]$_i$ of 0.05 µM (except for simulations with Ca$^{2+}$ levels below this basal value, for those we assumed [Ca$^{2+}$]$_{rest}$=[Ca$^{2+}$]$_i$). To compute the steady state, we assumed that no fusion took place, ignoring the very low fusion rate at resting [Ca$^{2+}$]$_i$. Under these conditions, the model is a closed system of recurrent states and obeys microscopic reversibility, that is for every closed loop state diagram, the product of the rate constants around the loop is the same in both directions (*Colquhoun et al., 2004*). Microscopic reversibility implies detailed balance, meaning that every reaction is in equilibrium at steady state. Thus, for any two states S$_i$ and S$_j$ which are connected by a reaction, the steady state distribution obeys

$$\left[S_j\right] = \frac{r_{ji}}{r_{ij}}\left[S_i\right]$$

where [S$_i$] and [S$_j$] are steady state quantities and r$_{ij}$ and r$_{ji}$ are the reaction rates between S$_i$ and S$_j$. Using this property, we calculated the steady state iteratively by setting the population of the first state (state (0,0,0)) to 1, and thereafter iteratively computing the population of the following state (following the lexicographic ordering as described above) using the following formulae:

$$\text{If } k > 0: \; \left[(n,m,k)\right] = \frac{r_{k-1,k}}{r_{k,k-1}}\left[(n,m,k-1)\right]$$

$$\text{If } m > 0 \text{ and } k = 0: \; \left[(n,m,0)\right] = \frac{r_{m-1,m}}{r_{m,m-1}}\left[(n,m-1,0)\right]$$

$$\text{If } n > 0 \text{ and } m = \; k = 0: \; \left[(n,0,0)\right] = \frac{r_{n-1,n}}{r_{n,n-1}}\left[(n-1,0,0)\right]$$

Afterwards, each state was divided by the sum of all state values and multiplied by the number of SVs in the RRP. In our model simulations, the size of the RRP was variable and followed a gamma distribution with a mean of 4000 SVs and a standard deviation of 2000 SVs (see *Figure 2—figure supplement 1*), based on experimental estimates from the calyx of Held (*Wölfel and Schneggenburger, 2003*). In the following calculations, we use $\varphi$ to denote the steady state probability vector (i.e. normalised to sum to 1).

## Computation of fusion probabilities and fusion rate

The analytical implementation of our model allowed us to compute the fusion rate and cumulative fusion probabilities with a constant [Ca$^{2+}$]$_i$ after stimulus onset (t=0), thereby mimicking conditions in Ca$^{2+}$ uncaging experiments. The constant [Ca$^{2+}$]$_i$ makes the model a homogenous Markov Model. The transition rates of the model can be organized in the intensity matrix, *Q*, such that,

$$Q_{i,j} = k_{i,j} \text{ for } i \neq j$$
$$Q_{i,i} = -\sum_{j \neq i} k_{i,j}$$

where $k_{i,j}$ is the rate of the reaction from state $i$ to state $j$. Given initial conditions, $\varphi$ (steady state normalized to a probability vector) and intensity matrix $Q_{\xi,C}$ ( $\xi$ being the free model parameters and C being the Ca²⁺ concentration),

$$p_{\xi,C}(t) = \varphi \exp(Q_{\xi,C}t)$$

is the distribution of SV states at time t, i.e. a *1 x $n_{states}$* vector with element $i$ being the probability of being in state $i$ at time point *t*. The single SV cumulative fusion probability (being a function, *G*, of time, *t*, defined by $\xi$ and C) is the last element, which we will denote with a subscript F,

$$G_{\xi,C}(t) = \left(p_{\xi,C}(t)\right)_F = \left(\varphi \exp(Q_{\xi,C}t)\right)_F \tag{1}$$

The fusion rate of a single SV can be calculated directly as the last element of the derivative of (**Equation 1**):

$$G'_{\xi,C}(t) = \left(\frac{dp_{\xi,C}(t)}{dt}\right)_F = \left(\varphi Q_{\xi,C} \exp(Q_{\xi,C}t)\right)_F \tag{2}$$

Multiplying (**Equation 1**) and (**Equation 2**) with the number of SVs yields the cumulative fusion function and fusion rate function, respectively. For simulation of release rates and release latencies (in **Figures 2–5**) and some figure supplements, we computed (**Equation 1**) and (**Equation 2**) using the best fit parameters from fitting with $M_{slots}$ = 3 and $n_{syts}$ = 15. This was done for 31 [Ca²⁺]ᵢ values ranging from 0.001 µM to 80 µM (0.001, 0.1, 0.2, …., 0.9, 1, 1.25, 1.5, 1.75, 2, 2.5, 3, 4, …, 9, 10, 20, 30, …, 80 µM) for $t \in [0,100]$ ms with a time step of 0.01ms. In addition, the functions were calculated in the same way using the best fit parameters for $M_{slots}$ = 1,2,4,5,6 with $n_{syts}$ = 15 for simulations depicted in (**Figure 2**) and figure supplemets. In some conditions, especially at low [Ca²⁺]ᵢ, a longer span of the cumulative fusion probability was required and was calculated with the same time step size.

## Computation of peak release rates

The peak of the fusion rate can be computed by multiplying the maximum value of the single SV fusion rate function, (**Equation 2**), with $n_{ves}$. To allow for a variable RRP size, a set of 1000 $n_{ves}$ values were drawn according to the RRP size distribution, the peak release rates were determined, and the mean and 95% prediction interval determined for each Ca²⁺ concentration.

For parameter exploration (**Figure 2G**) and for computing the release rates in the fitting routine, it was not feasible to calculate the fusion rate over 100ms with high temporal precision as described above. Instead, we implemented a custom search algorithm (scripts can be found in accompanying zip-file "Source_code1.zip"), which was constructed to shorten calculation time by taking advantage of the release rate function being unimodal. We first found a time point, $t_{max}$, at which 75–90% of the SVs had fused. Having computed different time points in the time interval *[0,$t_{max}$]*, gave us an interval in which the fusion rate showed a local maximum. The algorithm then narrowed the time interval down until a time of peak was found with a precision of 0.01ms. This method shortened simulation time considerably.

## Stochastic simulation of release latencies

Release latencies, which are defined as the time point of the fifth SV fusion event after the onset of simulation, were simulated stochastically by drawing $p_i \in (0,1)$, $i=1,…,n_{ves}$, from the uniform distribution for each of the 1000 repetitions and each evaluated [Ca²⁺]ᵢ. Each of these random numbers corresponds to the fusion time of an SV, which can be determined interpolation of the single SV cumulative fusion probability function (**Equation 1**). To obtain the time point of the fifth SV fusion, the fifth lowest $p_i$ was used. In the corresponding figures, the medians were plotted, since the probability distribution of the release latencies (derived below) was skewed, and the reported data points were single measurements.

## Fitting the model to experimental data

We next fitted the model to already published data describing the $Ca^{2+}$ dependence of NT release in the mouse calyx of Held (*Kochubey and Schneggenburger, 2011*). The data consist of measurements from $Ca^{2+}$ uncaging experiments, where the release latency, defined as the time point of the fifth SV fusion event, and the peak release rate were estimated at different $[Ca^{2+}]_i$. Besides the four free model parameters, $\xi$, an additional parameter, $d$, was fitted. $d$ is a constant added to the release latencies computed from the model to account for the experimentally observed delay (*Kochubey and Schneggenburger, 2011*).

Since the variance in the experimental data points also contains information on the underlying biological mechanism, we wanted to take the distribution of individual data points into account when obtaining estimates of the unknown parameters. We therefore derived the likelihood function, which describes how well the model captures the distribution of the release latencies. Obtaining this function for the peak release rates was not feasible. The experimental peak release rates were therefore compared to the average model prediction. Both measures of describing the correspondence between model simulations and experimental data were combined in a cost value which was optimized to estimate the best fit parameters (the lower this cost value the better the correspondence between model predictions and experimental data).

The best fit was obtained by minimizing the following cost function:

$$cost_{\xi,d} = 2 \cdot \sum_{i=1}^{n_{rates}} S_{\xi,i} - \sum_{j=1}^{n_{latencies}} \ell_{\xi,d,j}$$

where $i=1,\ldots, n_{rates}$ and $j=1,\ldots, n_{latencies}$ are the indices of the experimental $[Ca^{2+}]_i$,

$$S_{\xi,i} = \left(r_{max,\,model} - r_{max,experiment}\right)^2$$

are the squared deviations of the peak release rates ($1/ms^2$) and $\ell$ is the logarithm of the likelihood of the release latencies (see derivation below). To combine the two measures of distance between model and experimental data, the squared deviation of the peak release rates was multiplied by a factor 2 before subtracting the logarithm of the likelihood of the release latencies. The cost value was minimized using the inbuilt Matlab function *fminsearch,* which uses a Nelder-Mead simplex method. *fminsearch* was run with different initial parameter values to verify that the global minimum of the cost function was found. During the fitting, the lower and upper bounds of $d$ were set to, respectively, 0.3ms and 0.405ms (with the upper bound corresponding to the smallest release latency in the experimental data set). $\alpha,\gamma$, and $[PI(4,5)P_2]$ had an upper bound of $10^{10}$, and all free parameters needed to be positive. The maximum number of iterations and function evaluations was set to 5000.

## The likelihood function of release latencies with fixed RRP size

To fit the model to the experimental release latency measurements, we derived the likelihood function, which is the probability density function of the model for given parameters evaluated at the experimental data points. Thus, optimizing the likelihood function yields parameters for which the data points are most likely if the model is true. We first derive the likelihood of release latency in the case of a fixed RRP size ($n_{ves}$).

We define the stochastic variable $X$, which describes the stochastic process of the state of a single SV in our model. The fusion time of the SV, $\tau$, is defined as

$$\tau = \inf \{ t > 0 | X(t) = F \}$$

where $\tau$ itself is a stochastic variable. We define a stochastic vector, $Z$, which consists of all $n_{ves}$ fusion time points in a single experiment. They come from independent, identically distributed stochastic processes with cumulative distribution function $G_{\xi,C}(t)$, given in (*Equation 1*). As the release latency is defined as the time of the fifth SV fusion, we order the $Z$ variable outcomes ($z_{(1)}, z_{(2)}, \ldots, z_{(nves)}$) from first to last fusion time. Using the transformation

$$U_{(i)} = G_{\xi,C}\left(Z_{(i)}\right)$$

we obtain a sequence of stochastic variables, $U_{(i)}$, which are uniformly distributed on the interval (0,1). The ordering is preserved, since $G_{\xi,C}(t)$ is monotonically increasing, and $U$ has probability density function

$$f_{U_i}(t) = \left| G'_{\xi,C}\left( Z_{(i)} \right) \right|$$

with respect to $t$. $G_{\xi,C}'(t) \geq 0$ is given in (**Equation 2**). From order statistics it follows that the $k^{th}$ ordered $U$ is beta-distributed with probability density

$$f_{U_{(k)}}(u) = G'_{\xi,C}\left( G^{-1}_{\xi,C}(u) \right) \cdot \frac{n_{ves}!}{(k-1)!\,(n_{ves}-k)!} u^{k-1} \left( 1 - u \right)^{n_{ves}-k}$$

where $u = G_{\xi,C}(t)$. Thus, the transformed variable $U_{(k)}$, is beta-distributed, with

$$U_{(k)} \sim \text{Beta}\left( k,\ n_{ves} + 1 - k \right)$$

In the case of the release latency, we are interested in the fifth fusion event ($k=5$). Thus, with a fixed RRP size, the likelihood value for the release latency observations, $T^* = (t^*_1, t^*_2, ..., t^*_M)$, at all $M$ $Ca^{2+}$ concentrations is

$$\mathcal{L}_\xi\left( T^* \right) = \prod_{i=1}^{M} \left( G'_{\xi,C_i}\left( t^*_i \right) \frac{n_{ves}!}{4!\,(n_{ves}-5)!} G_{\xi,C_i}\left( t^*_i \right)^4 \left( 1 - G_{\xi,C_i}\left( t^*_i \right) \right)^{n_{ves}-5} \right)$$

In the optimization we minimize minus the log-likelihood:

$$\ell_\xi\left( T^* \right) = -\log\left( \mathcal{L}_\xi\left( T^* \right) \right) = -\sum_{i=1}^{M} \left( \log G'_{\xi,C_i}\left( t^*_i \right) \frac{n_{ves}!}{4!\,(n_{ves}-5)!} G_{\xi,C_i}\left( t^*_i \right)^4 \left( 1 - G_{\xi,C_i}\left( t^*_i \right) \right)^{n_{ves}-5} \right)$$

which is equivalent to maximizing the likelihood function.

## The likelihood of release latencies with variable RRP size

In our model, the RRP size is assumed to follow a Gamma distribution. Let $x$ denote the RRP size, $f_{RRP}(x)$ the probability density of the Gamma distribution, and $u = G_{\xi,C}(t)$ as defined above. The probability density of the release latency at $Ca^{2+}$ concentration $C_i$ is given by

$$
\begin{aligned}
\tilde{f}u_{(k)} &= \frac{1}{K} \int_5^\infty f_{RRP}(x)\, G'_{\xi,c_i}\left( G^{-1}_{\xi,c_i}(u) \right) \cdot \frac{x!}{4!(x-5)!} u^4 (1-u)^{x-5} dx \\
&= \frac{G'_{\xi,C}\left( G^{-1}_{\xi,C}(u) \right)}{K} \int_5^\infty f_{RRP}(x) \cdot \frac{x(x-1)(x-2)(x-3)(x-4)}{4!} u^4 e^{\log(1-u)x} e^{-5\log(1-u)} dx
\end{aligned}
\tag{3}
$$

where $K$ is a normalization constant, $K = 1 - P(x<5) \approx 1$. The lower limit of the integral reflects that the release latency is only defined when there are more than 5 SVs in the RRP. In simulations this corresponds to redrawing the RRP size whenever an RRP size <5 SVs occurs, which happens with probability ~3e-11, and is accounted for in the normalization constant K in the following. Inserting the probability density function of a Gamma distribution with shape parameter $k$ and scale parameter $\theta$, we get:

$$\tilde{f}u_{(k)} = \frac{G'_{\xi,C}\left( G^{-1}_{\xi,C}(u) \right)}{K} \int_5^\infty \frac{1}{\Gamma(k)\theta^k} \cdot x^{k-1} \cdot e^{\frac{-x}{\theta}} \cdot \frac{x(x-1)(x-2)(x-3)(x-4)}{4!} u^4 e^{\log(1-u)x} e^{-5\log(1-u)} dx$$

We now define the following variables:

$$\tilde{\theta} = \frac{\theta}{1 - \theta\log(1-u)} \ , \ \ a = \frac{u^4 e^{-5\log(1-u)} G'_{\xi,C}\left( G^{-1}_{\xi,C}(u) \right)}{4!\left( 1 - \theta\log(1-u) \right)^k} \tag{4}$$

By factoring out and substituting in the above equation we get

$$
\begin{aligned}
K\tilde{f}_{u_{(k)}}(u) &= \frac{G'_{\xi,C}\left(G^{-1}_{\xi,C}(u)\right)u^4 e^{-5(1-u)}}{4!(1-\theta\log(1-u))^k} \int_5^\infty \frac{(1-\theta\log(1-u))^k}{\Gamma(k)\theta^k} \cdot x^{k-1} \cdot e^{\frac{-(1-\theta\log(1-u))x}{\theta}} \\
&\quad \cdot x(x-1)(x-2)(x-3)(x-4)dx \\
&= a \int_5^\infty \frac{1}{\Gamma(k)\tilde{\theta}^k} \cdot x^{k-1} \cdot e^{\frac{-x}{\tilde{\theta}}} \cdot x\left(x-1\right)\left(x-2\right)\left(x-3\right)\left(x-4\right) dx
\end{aligned}
\tag{5}
$$

Furthermore, we have

$$
x\left(x-1\right)\left(x-2\right)\left(x-3\right)\left(x-4\right) = x^5 - 10x^4 + 35x^3 - 50x^2 + 24x
\tag{6}
$$

Since

$$
\Gamma\left(x+1\right) = x\Gamma\left(x\right)
$$

we can derive the following useful formula:

$$
\begin{aligned}
I_n &= \int_5^\infty \frac{1}{\Gamma(k)\tilde{\theta}^k} \cdot x^{k-1} \cdot e^{\frac{-x}{\tilde{\theta}}} \cdot x^n dx \\
&= \left(\prod_{m=0}^{n-1} k+m\right) \tilde{\theta}^n \int_5^\infty \frac{1}{\Gamma(k+n)\tilde{\theta}^{k+n}} \cdot x^{k+n-1} \cdot e^{\frac{-x}{\tilde{\theta}}} dx \\
&= \left(\prod_{m=0}^{n-1} k+m\right) \tilde{\theta}^n \left(1 - \frac{1}{\Gamma(k+n)}\gamma\left(k+n, \frac{5}{\tilde{\theta}}\right)\right)
\end{aligned}
\tag{7}
$$

The third equality follows from the fact that the function in the second integral from above is the probability density function of a gamma distribution with shape parameter $k+n$ and scale parameter $\tilde{\theta}$. We therefore replace it with the cumulative distribution function of the gamma distribution, where $\gamma$ is the lower incomplete gamma function. Combining (*Equations 4–7*) yields an explicit expression of the likelihood of a single delay in (*Equation 3*), as

$$
\tilde{f}_{U_{(k)}}\left(u\right) = \frac{a \cdot \left(I_5 - 10I_4 + 35I_3 - 50I_2 + 24I_1\right)}{K}
$$

with

$$
K = 1 - P\left(x < 5\right) = 1 - \frac{1}{\Gamma(k)}\gamma\left(k, \frac{5}{\tilde{\theta}}\right)
$$

We then minimize the sum of minus the log-likelihoods of the release latency observations.

## Syt binding states in the Gillespie simulation of model

In the Gillespie algorithm, the binding state of each individual syt is tracked. The state of the system at time point $t$, $X(t)$, is given by a $n_{syt}$ x $n_{ves}$ matrix. Each element in this matrix describes the binding state of an individual syt using a two-digit coding system; 00 for no species bound to syt, 01 for PI(4,5)$P_2$ bound, 10 for two $Ca^{2+}$ ions bound, and 11 for both species bound (dual-binding syt). As with the analytical implementation, each syt can undergo a subset of the 8 different (un)binding reactions (*Figure 1A*), depending on the binding state of the respective syt. The fusion rate, which depends on the number of dual-bound syts per SV, is determined for the entire SV.

## Determining the initial state of the system

The steady state (initial state, $X(0)$) was computed using the same method as described above (see section '*The steady state of the system*') using $[Ca^{2+}]_i = 0.05$ μM as the resting condition. This resulted in $\varphi$, the probability vector of a single SV to be in the different SV states at steady state. To stochastically determine $X(0)$, we first determined the binding state for each SV, that is how many dual bindings are formed ($n$) and how many syts have bound $Ca^{2+}$ ($m$) and how many PI(4,5)$P_2$ ($k$). For that we drew $p_j \in (0,1)$, $j=1\ldots,n_{ves}$, from the uniform distribution. The state number of the $j^{th}$ SV, $s$, was determined by:

$$
s = \text{smallest integer satisfying } \sum_{i=1}^{s} \varphi_i \geq p_j
$$

Via the ordering of states explained above, $s$ can be linked to the state triplet $(n_s, m_s, k_s)$. As the order of syts is irrelevant for model simulation, this information on the state of $SV_j$ was transferred to the $j^{th}$ column of the $X(0)$ matrix in a systematic way: The first $n_s$ elements were labeled with '11'; elements $n_s$ +1 to $n_s$ +$m_s$ were labeled with '10'; and elements $n_s$ +$m_s$ + 1 to $n_s$ +$m_s$ + $k_s$ were labeled with '01', and the remaining elements $(n_s$ +$m_s$ + $k_s$ +1) to $n_{syt}$ were set to '00'.

## Gillespie algorithm-based simulations of the model

For stochastic evaluation of the model by the Gillespie algorithm (*Gillespie, 2007*), we next introduced the propensity function $B$, which is defined by:

$B_{i,j}(x)\, dt$ : = the probability given $(X(t)=X)$, that the $i^{th}$ syt of the $j^{th}$ SV will undergo a reaction in the next infinitesimal time interval $[t, t+dt]$.

For element $i,j$ in $B$, the total reaction propensity is the sum of propensities of possible reactions given the binding state of the syt and can be computed as follow:

$$
B_{i,j}(X_{i,j}) =
\begin{cases}
\max\left(M_{slots} - PIP_{tot,j}, 0\right)\gamma\ \left[PI(4,5)P_2\right] + \alpha\left[Ca^{2+}\right]^2, & X_{i,j} = 00 \\
\delta + \alpha\left[Ca^{2+}\right]^2, & X_{i,j} = 01 \\
\beta + \max\left(M_{slots} - PIP_{tot,j}, 0\right)\gamma\ \left[PI(4,5)P_2\right], & X_{i,j} = 10 \\
A\delta + A\beta & X_{i,j} = 11
\end{cases}
$$

with $PIP_{tot,j}$ the total number of syts of SV $j$ bound to PI(4,5)P$_2$. To include the propensity for fusion of an SV, an additional row in $B$, which we index with the denotation $f$, describes the propensity of fusion for each SV in the matrix;

$$
B_{f,j} = l_+ f^{\sum_{i=1}^{n_{syt}} \mathbf{1}\left(X_{i,j}=11\right)}
$$

This makes $B$ a matrix of $(n_{syt}$ +1$)\times n_{ves}$. We denote the sum of all elements in $B$ by $B_0$. Using $B_0$ and 3 random numbers ($r_n \in (0,1)$, n=1,2,3) drawn from the uniform distribution, we determined the time step to the next reaction and which SV and syt this reaction affects. The time to next reaction, $\tilde{\tau}$, is given by

$$
\tilde{\tau} = \frac{1}{B_0}\ln\left(\frac{1}{r_1}\right)
$$

since it is exponentially distributed with rate $B_0$. The index, $j$, of the SV undergoing a reaction is the first index that satisfies:

$$
\sum_{j'=1}^{j}\sum_{i=1}^{n_{syt}+1} B_{i,j'} \geq r_2\, B_0
$$

Similarly, the index $i$ of the syt in SV $j$ undergoing a reaction is the smallest integer fulfilling:

$$
\sum_{i'=1}^{i} B_{i',j} \geq r_3 \sum_{\hat{i}=1}^{n_{syt}+1} B_{\hat{i},j}
$$

If the row index $i$ equals $n_{syt}$ +1, a fusion reaction occurs. The fusion time $(t + \tilde{\tau})$ is saved and all the propensities of SV $j$ in $B$ are set to 0. If $i$ is smaller or equal to $n_{syt}$ a binding or unbinding reaction of one of the two species occurs. To determine which of the four possible reactions is occurring, we define an additional propensity vector, $b_{react}$. The first element in $b_{react}$ denotes the propensity of PI(4,5)P$_2$ binding, the second element the propensity of Ca$^{2+}$ binding, and the third and fourth element the unbinding of PI(4,5)P$_2$ and Ca$^{2+}$, respectively. These elements are given by:

$$
b_{react,1} = \mathbf{1}_{\left(X_{i,j}=00\ \vee\ x_{i,j}=10\right)}\left(M_{slots} - PIP_{tot,j}\right)\gamma\ \left[PI(4,5)P_2\right]
$$

$$
b_{react,2} = \mathbf{1}_{\left(X_{i,j}=00\ \vee\ X_{i,j}=01\right)}\alpha\left[Ca^{2+}\right]^2
$$

$$
b_{react,3} = \mathbf{1}_{\left(X_{i,j}=01\ \vee\ X_{i,j}=11\right)}A^{\mathbf{1}\left(X_{i,j}=11\right)}\delta
$$

$$b_{react,4} = \mathbf{1}_{(X_{i,j}=10 \,\vee X_{i,j}=11)}\, A^{\mathbf{1}_{(X_{i,j}=11)}}\, \beta$$

Additionally, we define the transition matrix $V$, which describes the change in the state of $X_{i,j}$ induced by the four reactions:

$$V = \left(+01,\ +10,\ -01,\ -10\right)$$

A fourth random number, $r_4 \in (0,1)$, drawn from the uniform distribution, determines which reaction, $h$, occurs:

$$h = \text{smallest integer satisfying } \sum_{h'=1}^{h} b_{react,h'} \geq r_4 \sum_{\hat{h}=1}^{4} b_{react,\hat{h}}$$

The state of the corresponding SV and syt, $X_{i,j}$, is replaced by $X_{i,j} + V_h$ and t by $t + \tau$. Then $B$ is updated according to the change in $X$, and all steps are repeated. This iterative process continues until all SVs are fused, when simulating the model with a fixed $[Ca^{2+}]_i$.

When simulating AP-evoked responses (*Figures 5 and 6*), we used a $Ca^{2+}$ transient describing the microdomain $[Ca^{2+}]_i$ sensed locally by primed SVs in the mouse calyx of Held upon AP stimulation (*Wang et al., 2008*). This $Ca^{2+}$ transient also formed the basis for the $Ca^{2+}$ signal used to simulate a paired pulse stimulus (*Figure 7*), where the transients were placed with a 10 ms interval. Additionally, for the paired pulse stimulus, we added a residual $Ca^{2+}$ transient to the signal (exponential decay with amplitude: 0.4 µM, decay time constant: 0.154 s$^{-1}$). Similar to the uncaging simulations, the $[Ca^{2+}]_i$ before the onset of the stimulus was 0.05 µM. Since the $Ca^{2+}$ concentration is a factor in the reaction rates, the propensity matrices $B$ and $b_{react}$ were not only updated to the new state matrix, $X(t + \tilde{\tau})$, but also to a new $[Ca^{2+}]_i$ at time $t + \tilde{\tau}$. $[Ca^{2+}]_i$ at time $t + \tilde{\tau}$ was determined from the $Ca^{2+}$ transient using linear interpolation, and $B$ and $b_{react}$ were updated correspondingly. As the propensity of $Ca^{2+}$ binding is largely dependent on $[Ca^{2+}]_i$, the time step between two updates in $[Ca^{2+}]$ and the propensity matrices was set to be at most $8e^{-4}$s. If $\tilde{\tau}$ determined from $B$ at time $t$ was larger than this time step, no reaction occurred, but the system and $[Ca^{2+}]$ were updated to time $t+0.8$ms. The model was evaluated until the end of the $Ca^{2+}$ transient. Similarly, in the case of a variable $PI(4,5)P_2$ transient (*Figure 7*), the $PI(4,5)P_2$ concentration was updated at least every 0.8 ms. Because this approach requires simulation of all individual (un)binding reactions and fusion events it is not feasible to perform 1000 repetitions. Instead, simulations were repeated 200 times (*Figures 5 and 6*). Like with the computation of the release latencies and maximal fusion rates, we assumed a variable RRP. However, instead of drawing random pool sizes from a gamma distribution, we used the 200 quantiles of the pdf of the RRP sizes, because of the limited number of repetitions and the large impact of the RRP size on the model predictions. In *Figure 7*, we reduced the number of repetitions to 50, to represent an experimental condition more closely. The set of RRP sizes was drawn randomly once. This set of random RRP values was used for both the mutant and the WT condition displayed in the figure.

### Simulating the model with mutant syts

For mutations in syt that affect the binding and unbinding rates of $PI(4,5)P_2$ and $Ca^{2+}$, the procedure described above was repeated with adjusted parameters when simulating a model containing only mutant syts. For a model in which mutant proteins were expressed together with WT syts (simulations of heterozygous condition in *Figure 6*), the procedure was changed slightly.

For a model with $p$ WT and $q$ mutant syts, the number of states of an SV increases drastically and was now described by six values; the number of WT syts bound to $Ca^{2+}$, $PI(4,5)P_2$ or both, and the number of mutant syts bound to $Ca^{2+}$, $PI(4,5)P_2$ or both. The dimensions of the Q-matrix used to compute the steady-state probability of a single SV increased with $n_{states}$. $X(t=0)$ was computed using the same principle as described above, with the important difference that the first $p$ rows represented the binding status of the WT syts, and row $p+1$ to $p+q$ that of the mutants. In $B$ this ordering of WT and mutant syts is the same. The parameters used to compute $b_{react}$ depended on whether a reaction occurred to a WT syt, $i \leq p$, or a mutant syt, $n_{syt} \geq i > p$.

## Simulation of EPSCs

Simulated EPSCs were obtained using both model implementations. The analytical implementation of our model was used to simulate fusion times for a constant $[Ca^{2+}]_i$ (*Figure 4D and G*). The Gillespie version of the model was used to simulate AP-evoked EPSCs with or without mutant syts (*Figure 5C–E*, *Figure 5—figure supplement 2E-F*, *Figure 6C–D*, *Figure 6—figure supplement 1*, and *Figure 7B–D*). In both approaches, the stochastically determined fusion times, determined as described above, were rounded up to the next 0.02ms, leading to a sampling rate of 50 kHz. The sampled fusion times were convolved with a mEPSC to generate simulated EPSCs. The standard mEPSC used for deconvolution followed the equation described by *Neher and Sakaba, 2001*:

$$I_{mini}\left(t\right) = A \cdot \left(1 - \rho\right) \exp\left(-\frac{t}{\tau_1}\right) + \rho \cdot \exp\left(-\frac{t}{\tau_2}\right) - \exp\left(-\frac{t}{\tau_0}\right)$$

with $\tau_1$=0.12 ms (time constant of fast decay), $\tau_2$=13 ms (time constant of slow decay), $\tau_0$=0.12 ms (time constant of rise phase), $\rho$=1.e-5 (proportion of slow phase in decay), and $A$ being a normalization constant making the amplitude 60 pA. Parameter values were chosen such that the kinetics of the mEPSC would match events measured in the Calyx of Held (*Chang et al., 2015*). In *Figure 4D and G* traces show three randomly chosen eEPSCs in each panel. Representative eEPSC traces shown in *Figure 5*, *Figure 5—figure supplement 2*, *Figure 6*, *Figure 6—figure supplement 1*, and *Figure 7* are simulated eEPSCs with the amplitude closest to the mean eEPSC amplitude.

## Simulating AP-evoked EPSCs with variable number of syt

To investigate the effect of variability in the number of syts expressed per SV on variance between simulated AP-evoked traces (*Figure 5*), we first had to determine the steady state. For this we computed the probability vector of a single SV to be in the different SV-states at steady state ($\varphi$) for $n_{syt}$ = 1,…,50. Subsequently, for each SV and each repetition (n=1000) a random number of $n_{syt}$ was drawn from the Poisson distribution with mean = 15. When the value 0 was drawn, it was replaced by $n_{syt}$ = 1. Using these values and $\varphi$ determined for each $n_{syt}$, we computed the steady state matrix (X(0)) as described above ('Determining the initial state of the system'). To reduce computation time, we evaluated a model containing 100 vesicles 40 times instead of evaluating 4000 vesicles simultaneously. The fusion times obtained when driving the model with the $Ca^{2+}$ transient were combined afterwards. This is valid since all SVs act independently in the model. For the condition with a variable RRP size, fusion times were selected randomly until the RRP size of that specific repetition was reached. Afterwards, the fusion times were convolved with the mEPSC to obtain simulated AP-evoked responses. To quantify the variance in the traces (*Figure 5E*), we computed the standard deviation of the simulated eEPSCs at each data point (300 data points corresponding to the time interval 0–6ms) and summed those values.

## Acknowledgements

We thank Ralf Schneggenburger and Holger Taschenberger for providing the experimental $Ca^{2+}$ uncaging dataset and AP-induced $Ca^{2+}$ transient.

# Additional information

### Funding

| Funder | Grant reference number | Author |
| --- | --- | --- |
| Novo Nordisk Fonden | NNF19OC0056047 | Alexander M Walter |
| Novo Nordisk Fonden | NNF20OC0062958 | Susanne Ditlevsen |
| Novo Nordisk Fonden | NNF19OC0058298 | Jakob B Sørensen |
| Lundbeckfonden | R277-2018-802 | Jakob B Sørensen |
| Independent Research Fund Denmark | 8020-00228A | Jakob B Sørensen |

| Funder | Grant reference number | Author |
|---|---|---|
| Deutsche Forschungsgemeinschaft | 278001972 | Alexander M Walter |
| Deutsche Forschungsgemeinschaft | 261020751 | Alexander M Walter |

The funders had no role in study design, data collection and interpretation, or the decision to submit the work for publication.

### Author contributions

Janus RL Kobbersmed, Manon MM Berns, Formal analysis, Investigation, Visualization, Writing – original draft, Writing – review and editing; Susanne Ditlevsen, Jakob B Sørensen, Alexander M Walter, Conceptualization, Supervision, Funding acquisition, Writing – original draft, Writing – review and editing

### Author ORCIDs

Janus RL Kobbersmed http://orcid.org/0000-0003-0313-6205
Manon MM Berns http://orcid.org/0000-0003-2998-4202
Susanne Ditlevsen http://orcid.org/0000-0002-1998-2783
Jakob B Sørensen http://orcid.org/0000-0001-5465-3769
Alexander M Walter http://orcid.org/0000-0001-5646-4750

### Decision letter and Author response

Decision letter https://doi.org/10.7554/eLife.74810.sa1
Author response https://doi.org/10.7554/eLife.74810.sa2

## Additional files

### Supplementary files

• Transparent reporting form

• Source code 1. Source code for simulation, analysis and visualisation.

### Data availability

All data and software codes generated and used during this study are included in the manuscript and supporting files. Source data is included for all figures.

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
