## [Editor Report]

The calcium dependence of vesicle exocytosis at synapses is a power law with an exponent n = 3 or 4, however, the molecular mechanisms that underpin this highly non-linear dependence on calcium are unclear. To shed light on this fundamental question the authors build a model where 2 calcium ions bind to the protein synaptotagmin and synaptotagmin binds to the negatively charged lipid PIP2 in the presynaptic membrane. Simulations fit best the data from the calyx of Held synapse when 3 synaptotagmin molecules each bind calcium and PIP2. This compelling model shows that each Ca-synaptotagmin-PIP2 complex reduces the energy barrier for vesicle fusion by ~5k, thus, fast exocytosis at CNS synapses may require only 3 Ca-synaptogamin-PIP2 molecules to achieve submillisecond speeds of vesicle fusion.

---

## [Decision Letter]

**Decision letter after peer review:**

Thank you for submitting your article "Allosteric stabilization of calcium and lipid binding engages three synaptotagmins in fast exocytosis" for consideration by *eLife*. Your article has been reviewed by 3 peer reviewers, and the evaluation has been overseen by a Reviewing Editor and Richard Aldrich as the Senior Editor. The following individual involved in review of your submission has agreed to reveal their identity: Victor Matveev (Reviewer #1).

Essential revisions:

1) Lines 670-672: The main assumption of the model is that near-simultaneous binding of 2 calcium ions greatly increases the affinity of PIP2 binding. However, simultaneous binding of 2 Ca^2+^ ions must imply an even stronger cooperativity between the two Ca^2+^ binding events than the cooperativity between Ca^2+^ and PIP2 binding. While equilibrium properties are not affected by such an assumption, the time-dependence of the model should depend on the precise sequence of binding events. Would the release latency predictions change if both Ca^2+^ binding events were resolved? Of course, the stochastic simulations for a more complicated model would be costly, and extra model parameters are not desired, but one trial analytic solution for a constant Ca^2+^ level is probably not hard. I am not insisting that any new simulations be included in the paper, but it would help to discuss briefly whether the assumption of two simultaneous Ca^2+^ binding affects the release latency estimation.

2) On page 10, the Authors describe their checks to ensure that the model parameter optimization procedure is not stuck at a local minimum. However, a more straightforward common check would involve repeating the fminsearch parameter optimization algorithm multiple times, while starting at different initial values of parameters. Have the Authors performed such a check? This would essentially quantify uncertainties in inferred parameter values in Table 2.

3) In the discussion (Line 542) the authors note that their model is fundamentally different from the proposed preassembly of Syt rings at the base of a vesicle, and that this is big testable prediction. How could this be tested? And do Syts actually preassemble with slots in their model? Figure 3B shows the number of crosslinks that precede fusion, and it would be nice to see how many Syts are bound to PIP2 at rest for any given number of "slots".

4) I like Figure 2 Sup2, where the authors probe the number of crosslinks prior to fusion. Interestingly, the speed of Ca ramping affects the number of crosslinks, and a significant number of events occur after 4 crosslinks only when Ca increases at the fastest rate (0.001 s). This is still much slower than the expected rate of Ca nanodomain formation (<100 us). Why did the authors explore only extremely slow Ca ramp rates? It seems that the ramping rates of 10 us to 10 ms would more accurately affect the condition of action potential induced Ca influx.

5) It is not immediately clear to me why the model with 6 slots never predicts fusion with 5-6 crosslinks. Why does the model fail to fit well if each crosslink contributes less to lowering the energy barrier, and thus 5-6 crosslinks are required to drive fusion?

6) If a single crosslink brings the vesicle closer to the PM, this will introduce a new form of allostery by increasing the effective concentration of PIP2 sensed by Syts. Is this reflected in the model?

7) In the model, PIP2 binding is required for Syt1-driven fusion. As a result, only Syts with a "slot" can participate in fusion. However, while K325, 327A mutations that disrupt Ca-independent PIP2 binding lead to less synchronous release, synchronous release can be restored using a paired-pulse stimulation paradigm where an initial action potential drives Ca-dependent membrane attachment of vesicles to the plasma membrane. This suggests that PIP2 binding is not required for "cross linking". How might the model change if Syts without slots could also contribute to fusion in your model?

8) The evaluation of mutants in Figure 5 should be better tied to actual biology. What mutation does "Ca^2+^ binding", and "A-on" mimic? No citations are provided for the many studies where Syt mutants were expressed in Syt1 KO neurons. This seems like a perfect place to test the role of PIP2 binding with more complicated Ca stimuli. Could the model be adapted to explore the effect of K/A mutations described by Chang, Trimbuch and Rosenmund?

9) Recent experimental and modeling work with Synaptotagmin and SNAREs has been published (Wu et al., 2021; *eLife*; https://elifesciences.org/articles/68215). These authors say: "To test whether Syt1 affected fusion pores in this system, we co-reconstituted ~4 copies of recombinant full-length Syt1 together with ~4 copies of VAMP2 (per disc face) into large nanodiscs called nanolipoprotein particles…". The work seems particularly relevant to your paper. Also please take a look at another paper from this group that suggests fast exocytosis requires up to 15 SNAREs and Syt complexes (Wu et al., *eLife*, 2017; https://elifesciences.org/articles/22964). Please read and discuss in your paper these recent modeling studies, which seem to suggest that large numbers of Syt's and SNAREs are needed for fast exocytosis at synapses.

*Reviewer #1 (Recommendations for the authors):*

1) Lines 670-672: The main assumption of the model is that near-simultaneous binding of 2 calcium ions greatly increases the affinity of PIP2 binding. However, simultaneous binding of 2 Ca^2+^ ions must imply an even stronger cooperativity between the two Ca^2+^ binding events than the cooperativity between Ca^2+^ and PIP2 binding. While equilibrium properties are not affected by such an assumption, the time-dependence of the model should depend on the precise sequence of binding events. Would the release latency predictions change if both Ca^2+^ binding events were resolved? Of course, the stochastic simulations for a more complicated model would be costly, and extra model parameters are not desired, but one trial analytic solution for a constant Ca^2+^ level is probably not hard. I am not insisting that any new simulations be included in the paper, but it would help to discuss briefly whether the assumption of two simultaneous Ca^2+^ binding affects the release latency estimation.

2) On page 10, the Authors describe their checks to ensure that the model parameter optimization procedure is not stuck at a local minimum. However, a more straightforward common check would involve repeating the fminsearch parameter optimization algorithm multiple times, while starting at different initial values of parameters. Have the Authors performed such a check? This would essentially quantify uncertainties in inferred parameter values in Table 2.

3) Apart from the nice connection between the proposed model and the model of Lou, Scheuss and Schneggenburger (2005), I think the presented model can also be viewed as a more detailed and biophysically-based extension of the "excess-calcium binding site model" of S.D. Meriney and coworkers, which I would recommend citing. To my knowledge, there are 3 papers that make use of the latter model, in one form or another (but I would recommend that the Authors double-check these papers to see if all of them are relevant):

Dittrich M, Pattillo JM, King JD, Cho S, Stiles JR, Meriney SD (2013) An excess-calcium binding site model predicts neurotransmitter release at the neuromuscular junction. Biophys J 104: 2751-63

Ma J, Kelly L, Ingram J, Price TJ, Meriney SD, Dittrich M (2015) New insights into short-term synaptic facilitation at the frog neuromuscular junction. J Neurophysiol 113: 71-87

Luo F, Dittrich M, Cho S, Stiles JR, Meriney SD (2015) Transmitter release is evoked with low probability predominately by calcium flux through single channel openings at the frog neuromuscular junction. J Neurophysiol 113: 2480-9

*Reviewer #2 (Recommendations for the authors):*

1. The style and length of this paper resemble a thesis. The authors are encouraged to edit the manuscript to make it accessible to a wide *eLife* audience. Many figures have complicated subpanels that are difficult to understand, and the text is often so technical that it does not convey the essential point of each argument. This is a very nice model. The authors could do a better job of describing the important points, which they highlight most clearly at the beginning of the discussion.

2. The model depicted in Figure 1A suggests that Ca-bound C2B attaches to the vesicle, rather than the PM. However, Chang, Trimbuch and Rosenmund 2019 showed that the PIP2 binding attaches vesicles to the PM before an action potential. Do the authors want to claim that Ca-bound C2B attaches to vesicles? This is actually very important to the main point of the paper, because "crosslinks" are likely to occur in the absence of Ca, and Ca-independent PIP2 binding might position the Ca-binding pocket closer to negatively charged phospholipids, increasing the affinity of C2B for Ca.

3. In the discussion (Line 542) the authors note that their model is fundamentally different from the proposed preassembly of Syt rings at the base of a vesicle, and that this is big testable prediction. How could this be tested? And do Syts actually preassemble with slots in their model? Figure 3B shows the number of crosslinks that precede fusion, and it would be nice to see how many Syts are bound to PIP2 at rest for any given number of "slots".

4. Line 149: What (little) is known about the concentration of PIP2 at active zones? How does the concentration of PIP2 in rich patches compare to the idea that only 3 PIP2 molecules could be available beneath the space of one vesicle? I found the discussion of PIP2 concentration in Methods (Line 999) difficult to understand.

5. The model only accounts for Ca binding by the C2B domain, which as the authors note is the more important of the C2 domains. C2A is barely discussed in this paper. However, there are differences in fusion rates when the C2A domain is mutated to block Ca binding. How would the addition of C2A affect the model?

6. I like Figure 2 Sup2, where the authors probe the number of crosslinks prior to fusion. Interestingly, the speed of Ca ramping affects the number of crosslinks, and a significant number of events occur after 4 crosslinks only when Ca increases at the fastest rate (0.001 s). This is still much slower than the expected rate of Ca nanodomain formation (<100 us). Why did the authors explore only extremely slow Ca ramp rates? It seems that the ramping rates of 10 us to 10 ms would more accurately affect the condition of action potential induced Ca influx.

7. It is not immediately clear to me why the model with 6 slots never predicts fusion with 5-6 crosslinks. Why does the model fail to fit well if each crosslink contributes less to lowering the energy barrier, and thus 5-6 crosslinks are required to drive fusion?

8. If a single crosslink brings the vesicle closer to the PM, this will introduce a new form of allostery by increasing the effective concentration of PIP2 sensed by Syts. Is this reflected in the model?

9. In the model, PIP2 binding is required for Syt1-driven fusion. As a result, only Syts with a "slot" can participate in fusion. However, while K325, 327A mutations that disrupt Ca-independent PIP2 binding lead to less synchronous release, synchronous release can be restored using a paired-pulse stimulation paradigm where an initial action potential drives Ca-dependent membrane attachment of vesicles to the plasma membrane. This suggests that PIP2 binding is not required for "cross linking". How might the model change if Syts without slots could also contribute to fusion in your model?

10. The evaluation of mutants in Figure 5 should be better tied to actual biology. What mutation does "Ca^2+^ binding", and "A-on" mimic? No citations are provided for the many studies where Syt mutants were expressed in Syt1 KO neurons. This seems like a perfect place to test the role of PIP2 binding with more complicated Ca stimuli. Could the model be adapted to explore the effect of K/A mutations described by Chang, Trimbuch and Rosenmund?

*Reviewer #3 (Recommendations for the authors):*

Such kind of complex models tend to have various underlying assumptions which could significantly influence the conclusions. Three examples are provided:

1. The interaction of synaptotagmins with SNARES or other proteins could change the Ca^2+^ affinity but only PIP2 is considered as a potential interaction for changing the Ca^2+^ affinity. In other word, the premise of the paper that the Ca^2+^ affinity of synaptotagmin in vitro and in vivo (within the protein complex of the fusion machinery) is identical could be wrong. Therefore, the main conclusion of the strong cooperativity of Ca^2+^ and PIP2 could be wrong.

2. The assumption that each crosslinking synaptotagmin lowers the energy barrier for fusion by the same amount (E_syt) could be wrong. Assuming a positive or negative cooperativity in E_syt per synaptotagmin might impact the conclusions regarding the number of required synaptotagmins.

3. The distance between the individual synaptotagmin molecules and the nearest Ca^2+^ channels could differ significantly on the nm-scale. The assumption of an identical local Ca^2+^ signal for all synaptotagmins could be wrong which could complicate the model predictions of the effect of reduced synaptotagmin copy numbers (Figure 4 and 5).

In conclusion, the study provides interesting and plausible possibilities of how synaptotagmins could mediate vesicle fusion but experimental validations are required to increase the amount of reliable and novel insights.

---

## [Author Response]

Essential revisions:1) Lines 670-672: The main assumption of the model is that near-simultaneous binding of 2 calcium ions greatly increases the affinity of PIP2 binding. However, simultaneous binding of 2 Ca^2+^ ions must imply an even stronger cooperativity between the two Ca^2+^ binding events than the cooperativity between Ca^2+^ and PIP2 binding. While equilibrium properties are not affected by such an assumption, the time-dependence of the model should depend on the precise sequence of binding events. Would the release latency predictions change if both Ca^2+^ binding events were resolved? Of course, the stochastic simulations for a more complicated model would be costly, and extra model parameters are not desired, but one trial analytic solution for a constant Ca^2+^ level is probably not hard. I am not insisting that any new simulations be included in the paper, but it would help to discuss briefly whether the assumption of two simultaneous Ca^2+^ binding affects the release latency estimation.

In our model, the binding of the two Ca^2+^ ions is described in one reaction, and the biological implications are connected to the state in which both Ca^2+^ ions have bound. This generates a simple interpretation regarding both the allosteric effect and the effect the C2B domains exert on the energy barrier for fusion:

1. The allosteric coupling between Ca^2+^ and PI(4,5)P_2_ is active when the Ca^2+^ binding sites are saturated (otherwise it is inactive).

2. Whenever a C2B domain binds both Ca^2+^ and PI(4,5)P_2_ the energy barrier for fusion is lowered.

If we split up the binding of Ca^2+^ into two consecutive reactions, we are faced with the problem of how to assign these two effects. For instance, is the allosteric effect “divided” between the Ca^2+^ bindings, or is it in effect only when both ions are bound? Does association of one Ca^2+^ together with PI(4,5)P_2_ have a partial effect on the energy barrier for fusion? How are the effects split up? As we do not know the answers to these questions, we circumvented these by assuming a simpler case in which both Ca^2+^ ions needed to be bound to exert these effects.

We agree with the reviewer that the exact mechanism is likely more complex. The coordination of the Ca^2+^ ions likely induces some conformational changes (e.g. membrane insertion) that stabilize the Ca^2+^ association and when this happens upon binding a single Ca^2+^ ion, the affinity for binding a second ion would likely be much higher. By simplifying the two binding reactions to one, we indeed indirectly assumed a high cooperativity between the binding events. We now added a statement of this to the methods section of the manuscript.

2) On page 10, the Authors describe their checks to ensure that the model parameter optimization procedure is not stuck at a local minimum. However, a more straightforward common check would involve repeating the fminsearch parameter optimization algorithm multiple times, while starting at different initial values of parameters. Have the Authors performed such a check? This would essentially quantify uncertainties in inferred parameter values in Table 2.

Repeating fminsearch with different initial parameters is a common way to confirm detection of the global minimum. Indeed, we had performed such an approach initially and have now also added a more systematic investigation. However, in our manuscript, we aimed to give more insight in the optimization space and therefore presented the analyses shown in figure 2F and 2G. These analyses show that our optimization problem has (within the parameter space that we explored) a global minimum at a cost of -139.45 and a local minimum with a cost of -107.

To address the reviewer’s comment we now fitted the model with 100 random initial values in which α, γ and F varied between 1 and 1000, [PI(4,5)P_2_] between 1 and 100. The initial value for the added delay was set to 0.4 ms, as we approximately know its value. 40 out of 100 fits ended in the reported global minimum (cost = -139.45). When running the optimization procedure a second time with the fitted parameters as new initial values, a total of 60/100 fits ended in the reported minimum (-139.45). During this fitting procedure, we also found a total of 18 fits with cost = -107.15 and a fitted f-value of 989. Given that we observe a local minimum in figure 2H at f~1000, it is not surprising that multiple fits also converged to this minimum. The remaining 22 fits are likely not to have converged yet. Thus, running the fminsearch multiple times shows that the analysis performed in figure 2G and 2H, gives a good insight in the optimization problem.

3) In the discussion (Line 542) the authors note that their model is fundamentally different from the proposed preassembly of Syt rings at the base of a vesicle, and that this is big testable prediction. How could this be tested? And do Syts actually preassemble with slots in their model? Figure 3B shows the number of crosslinks that precede fusion, and it would be nice to see how many Syts are bound to PIP2 at rest for any given number of "slots".

The model of the preassembled syt rings (see for example Rothman et al., 2017) implies that rather specific quantities of synaptic proteins are needed to induce fusion. Such a model would be highly sensitive to reducing the number of participating syts. Transmitter release would immediately break down if the number of syts were reduced such that no rings could preassemble. This contrasts our model where titration of syt copy number per SV yields a gradual effect on release and where a high number of syt copies is not needed for SV fusion per se but ensures fast and reliable neurotransmission by increasing the speed of the response to a stimulus (Figure 5). This difference suggests a way to test these hypotheses experimentally. We extended our Discussion section on this.

Whether syts preassemble with slots/PI(4,5)P_2_ in our model is a very interesting question. We therefore added the results on the steady state distribution as an additional figure (figure 3, Figure 3 —figure supplement 1) to our manuscript. Besides looking into the steady state distribution, we also investigated whether those vesicles that have multiple syts associated to PI(4,5)P_2_ have a higher probability of contributing to fusion. Below, we describe these results.

The histogram in Author response image 1 illustrates the steady state for each choice of M_slots_ (see also Figure 3A, and figure 3 —figure supplement 1A for steady state distributions of M_slots_ = 3 and M_slots_ = 6 respectively). For each choice of M_slots_, we used the corresponding best fit parameters, which makes parameters vary between choices of M_slots._ The bars show the percentage of SVs having bound any number of PI(4,5)P_2_ molecules at steady state without forming any crosslinks. Formation of a crosslink at steady state happens with probability between 0.02 % and 0.0001 % depending on the number of slots.

**Author response image 1. sa2fig1:** 

This shows that for M_slots_≥3 a considerable number of SVs preassembles with PI(4,5)P_2_ prior to the stimulus. In the following plots we investigated the consequence of steady state PI(4,5)P_2_ binding for the best fit situation, that is for M_slots_=3.An SV with one or more syts having bound PI(4,5)P_2_ at steady state naturally can have an “advantage” in release probability. This is illustrated in Author response image 2, where the release probability of SVs with a certain amount of PI(4,5)P_2_ bound at steady state is plotted against time after a Ca^2+^ flash of 50 µM see also Figure 3C (M_slots_ = 3), and Figure 3—figure supplement 1C (M_slots_=6).

Combining the steady state amounts with the release probabilities, we can compute the cumulative number of SV fusions over time with an RRP size of 4000 SVs (Figure 3B).Relatively many of the fastest SV fusions are a result of syts preassembling with PI(4,5)P_2_. Restricting the plot to the first 0.05 ms, we see that preassembling with PI(4,5)P_2_ really affects the initial response to this stimulus, which is highly relevant for the obtained release latency (defined as the time between stimulus and the 5^th^ SV fusion event).

Since the affinities for Ca^2+^ and PI(4,5)P_2_ are fixed to experimental values, the PI(4,5)P_2_ concentration is the only fitted parameter that can actually change the PI(4,5)P_2_ binding status at steady state. This provides an explanation for the large impact of the PI(4,5)P_2_ concentration on the release latencies (as seen in Figure 2 —figure supplement 5) and justifies the fitting of both the PI(4,5)P_2_ concentration and the PI(4,5)P_2_ binding rate, as these parameters affect the model in very different ways.

The above analyses are performed on the level of the vesicles. We also wanted to investigate the relevance of having bound to PI(4,5)P_2_ at steady state at the level of individual syts.

For this, like in some parts of the manuscript, we used the Gillespie algorithm. We ran the algorithm with 100 vesicles 200 repetitions for the Calcium steps to 1, 5, 10, 50, and 100 µM. As we were interested in investigating the probability of being involved in fusion (engaging in Ca^2+^/PI(4,5)P_2_ dual-binding) for those syts that have prebound to PI(4,5)P_2_ at steady state, we only ran the algorithm until all syts that were bound to PI(4,5)P_2_ at steady state had either fused or un-bound PI(4,5)P_2_. In Author response table 1 you can find the proportion of syts prebound to PI(4,5)P_2_ at steady state that is involved in fusion for the different Calcium concentrations analysed. In line with the above results, at a calcium concentration of 50 µM, the vast majority of the syts that have bound PI(4,5)P_2_ at steady state are also actively contributing to fusion by dual binding Ca^2+^/PI(4,5)P_2_. The proportion of prebounds syts involved in fusion is highly dependent on the Calcium concentration. At low Calcium concentrations (1 and 5 µM) having PI(4,5)P_2_ bound at steady state is not a predictor for being involved infusion.

**Author response table 1. sa2table1:** 

Calcium concentration (µM)	Fraction of PI(4,5)P_2_-prebound Syts that engage in Ca^2+^/PI(4,5)P_2_ dual-binding during fusion
1	3.876e-5
5	0.0479
10	0.3319
50	0.9439
100	0.9850

4) I like Figure 2 Sup2, where the authors probe the number of crosslinks prior to fusion. Interestingly, the speed of Ca ramping affects the number of crosslinks, and a significant number of events occur after 4 crosslinks only when Ca increases at the fastest rate (0.001 s). This is still much slower than the expected rate of Ca nanodomain formation (<100 us). Why did the authors explore only extremely slow Ca ramp rates? It seems that the ramping rates of 10 us to 10 ms would more accurately affect the condition of action potential induced Ca influx.

This relies on a misunderstanding. We did explore very fast (i.e. instantaneous behaviour) in the left hand column and then explored slower ramps in the right hand column. At the fastest rise time (0.001 s), convergence was already reached, as the average of approx. 3.4 crosslinks before fusion was also seen with 100 µM step Ca^2+^. We have now tried to make this point clearer by adding a few faster rise time points, making the x-axis logarithmic as well as drawing a horizontal line that visualises the convergence.

5) It is not immediately clear to me why the model with 6 slots never predicts fusion with 5-6 crosslinks. Why does the model fail to fit well if each crosslink contributes less to lowering the energy barrier, and thus 5-6 crosslinks are required to drive fusion?

Indeed, it is striking that we consistently see 3-4 dual bindings formed before fusions – even when allowing up to 6 dual-bounds. Fitting the model with different choices of M_slots_≥3 yielded consistent estimation of the contribution to the energy barrier for fusion by each syt engaging in Ca^2+^/PI(4,5)P_2_ dual binding (Figure 2F), even when probing many different initial parameter values in the fitting routine. As the reviewer notes, if each individual dual-binding syt contributed less to the energy barrier for fusion, more C2B domains would be required to engage simultaneously to achieve the fast rates of SV fusion. However, engaging more domains also would require the binding of more Ca^2+^ ions which could alter the Ca^2+^ dependence of the release rates. To investigate this, we ran the fitting procedure forcing the model with 6 slots to induce fusion with fastest rate only after 6 dual-bounds have been formed. This we did by fixing the contribution of each individual dual-binding syt to the energy barrier (factor f) during the fitting procedure and only fitted the four remaining parameters for a model with M_slots_ = 6. We computed the fixed f value based on the f value obtained by fitting all five model parameters, which equalled 163.5 for the six slots condition (M_slots_ = 6). The f value that, under the same conditions, would lead to most fusion events to occur once 5/6 dual-bounds have formed can be computed by: (163.5)36=12.79 (see Figure 2 —figure supplement 4 for confirmation). The best fit that could be obtained using f = 12.79 had a cost value of 8.78, which is much larger than the best fit obtained by fitting with f as a free parameter (cost = -126.5). Figure 2 —figure supplement 4 (panel A) show the best fit release latencies and peak release rates when forcing the six syts engaging in Ca^2+^/PI(4,5)P_2_ dual binding during fast fusion. The 95% prediction interval captures less of the data points for both the release latency and peak release rates condition compared to the original fit. Furthermore, as expected, the model predicted a calcium-dependency of the peak release rates which was too steep compared to the experimental data. Panel B shows that with these new settings, most fusion events occur when 6 crosslinks have formed. We added this figure as an additional supplement to our manuscript (Figure 2 —figure supplement 4)

The requirement for three crosslinking syts for fusion likely relates to the assumption that each C2B domain binds two Ca^2+^ ions. We also tested whether the condition with six slots and the binding of a single Ca^2+^ ion to each synaptotagmin could result in model predictions corresponding to the experimental data (Ca^2+^ affinity and allostericity value adjusted such that they correspond to binding of a single Ca^2+^ ion), and show fusion from a condition were six synaptotagmins are binding both Ca^2+^ and PI(4,5)P_2_. Also with these settings (using K_D,Ca_^2+^ = 221 µM, and A = 0.0015), the model was not fitting to the experimental data (see Author response image 3 and Author response table 2).

**Author response table 2. sa2table2:** 

Parameter	M_slots_ = 6, original fit	M_slots_ = 6, Fixed F	M_slots_ = 6, single Ca^2+^ binding
*α* (µM^-2^s^-1^)	24.11	22.96	-
*α_single_* (µM^-1^s^-1^)	-	-	30.86
*γ* (µM^-1^s^-1^)	126.6	0.49	1.56e9
[PI(4,5)P_2_] (µM)	0.2320	12.71	0.85
*f*(E_syt_ (K_b_T))	163.5(5.10)	12.79	17.25
*d*, added delay (ms)	0.3881	0.3850	0.33
**Costs**	-126.5	8.778	3.52

**Author response image 3. sa2fig3:** 

6) If a single crosslink brings the vesicle closer to the PM, this will introduce a new form of allostery by increasing the effective concentration of PIP2 sensed by Syts. Is this reflected in the model?

This is a very good point and we have added a new Figure (New Figure 7) to indicate the strong effect changes of the effective PI(4,5)P_2_ concentration has on synaptic responses (see point below). Currently our model does not include any heterogeneity in the effective PI(4,5)P_2_ concentration individual syts experience. This could be the case if all RRP vesicles are similarly docked. Indeed, our model predicts that most RRP SVs pre-associate PI(4,5)P_2_ (New Figure 3). However, it remains possible that binding of additional SV syts to PI(4,5)P_2_ further affects the effective PI(4,5)P_2_ concentration sensed by the remaining syts. We currently do not see how we might quantitatively estimate this effect. As a matter of fact, the effect may be bi-directional for the remaining syts on SV (e.g. while the effective PI(4,5)P_2_ concentration may increase for syts facing same PI(4,5)P_2_-patch on the plasma membrane, it may decrease for ones facing away from the PI(4,5)P_2_ patch). We currently do not know how the size of these effects may be estimated (this would require taking into account geometrical information, diffusion of the syt transmembrane anchor, protein lengths and flexibility, and the organization of PI(4,5)P_2_ in the membrane). We therefore kept our model as simple as possible in this regard, by assuming that all syst see the same PI(4,5)P_2_ concentration. Thus, the best fit PI(4,5)P_2_ concentration may reflect the average conditions for each syt. We now explain this clearer in the Results and touch on the above limitation in the Discussion.

7) In the model, PIP2 binding is required for Syt1-driven fusion. As a result, only Syts with a "slot" can participate in fusion. However, while K325, 327A mutations that disrupt Ca-independent PIP2 binding lead to less synchronous release, synchronous release can be restored using a paired-pulse stimulation paradigm where an initial action potential drives Ca-dependent membrane attachment of vesicles to the plasma membrane. This suggests that PIP2 binding is not required for "cross linking". How might the model change if Syts without slots could also contribute to fusion in your model?

To address this point, we performed additional simulations and have now added a new figure (Figure 7) to the manuscript where we explore the consequences of activity dependent SV repositioning. This point also nicely links to the one above regarding the effective PI(4,5)P_2_ concentration. As pointed out above, our model features one PI(4,5)P_2_ concentration that determines the likelihood that syts associate. This PI(4,5)P_2_ concentration encompasses both the density of PI(4,5)P_2_ in the membrane and the accessibility of syt to PI(4,5)P_2_. The latter will depend on how close the vesicle is docked to the plasma membrane. In their paper, Chang, Timbruch and Rosenmund (Chang et al., 2018) show that mutation of basic residues of the C2B domain (lysines also implicated in PI(4,5)P_2_ binding or the aspartates implicated in membrane binding) leads to loss of SV docking. Docking is restored upon AP-stimulation of mutant synapses and probing synaptic responses with a second AP 10 ms after the first leads to a large facilitation of synaptic responses. We here simulated such a condition by implementing a time-dependent increase in the effective PI(4,5)P_2_ concentration in such mutants. The assumption here is that when SVs are distant from the PM, the effective PI(4,5)P_2_ level available for syts are low (“the membrane is more difficult to reach”) but this situation normalizes upon SV translocation to the PM (Figure 7A). Accordingly, responses to the first AP are strongly reduced, but normalize for a second AP, causing strong short-term facilitation (Figure 7B-D). While our model clearly is a strong simplification (as stated above, the effective PI(4,5)P_2_ concentration in reality is a property relevant for each individual syt and this situation becomes more relevant here due to the more inhomogeneous distribution of RRP SVs) our model can provide a simple interpretation of the phenotypic cause of this type of mutations. We thank the reviewer for giving us the opportunity to explore this interesting point.

8) The evaluation of mutants in Figure 5 should be better tied to actual biology. What mutation does "Ca^2+^ binding", and "A-on" mimic? No citations are provided for the many studies where Syt mutants were expressed in Syt1 KO neurons. This seems like a perfect place to test the role of PIP2 binding with more complicated Ca stimuli. Could the model be adapted to explore the effect of K/A mutations described by Chang, Trimbuch and Rosenmund?

We agree that this section of the manuscript was not presented well. We have now rewritten the text dealing with the putative mutations that interfere with Ca^2+^ binding and the allosteric coupling between the Ca^2+^ and PI(4,5)P_2_ binding sites. We now specifically name and reference the mutations that our hypothetical model parameter changes may relate to. We have furthermore reduced the complexity of this figure (removed superfluous panels/supplementary items).

Following the reviewers’ suggestions (see point above) we have now also added an exploration of putative effects caused by mutations that interfere with plasma membrane association. In this context we also used the opportunity to provide further analysis on repetitive AP stimuli. The new results are added in a new section of the main text and one new main figure (Figure 7). Our simulations show that activity-dependent changes in the effective PI(4,5)P_2_ concentration (caused by SV repositioning) will have prominent effects on short-term synaptic plasticity. We thank the reviewer for this suggestion and for giving us the opportunity to provide these further insights.

9) Recent experimental and modeling work with Synaptotagmin and SNAREs has been published (Wu et al., 2021; eLife; https://elifesciences.org/articles/68215). These authors say: "To test whether Syt1 affected fusion pores in this system, we co-reconstituted ~4 copies of recombinant full-length Syt1 together with ~4 copies of VAMP2 (per disc face) into large nanodiscs called nanolipoprotein particles…". The work seems particularly relevant to your paper. Also please take a look at another paper from this group that suggests fast exocytosis requires up to 15 SNAREs and Syt complexes (Wu et al., eLife, 2017; https://elifesciences.org/articles/22964). Please read and discuss in your paper these recent modeling studies, which seem to suggest that large numbers of Syt's and SNAREs are needed for fast exocytosis at synapses.

We would like to thank the reviewer for pointing out these relevant papers. We have added the points to the discussion. We especially discuss that synaptotagmin was found to only change the fusion kinetics when PI(4,5)P_2_ was present and we included this finding in our manuscript to give a better explanation for why in our model only if both Ca^2+^ and PI(4,5)P_2_ are bound to the C2B domain the fusion barrier for fusion is reduced.

Reviewer #1 (Recommendations for the authors):1) Lines 670-672: The main assumption of the model is that near-simultaneous binding of 2 calcium ions greatly increases the affinity of PIP2 binding. However, simultaneous binding of 2 Ca^2+^ ions must imply an even stronger cooperativity between the two Ca^2+^ binding events than the cooperativity between Ca^2+^ and PIP2 binding. While equilibrium properties are not affected by such an assumption, the time-dependence of the model should depend on the precise sequence of binding events. Would the release latency predictions change if both Ca^2+^ binding events were resolved? Of course, the stochastic simulations for a more complicated model would be costly, and extra model parameters are not desired, but one trial analytic solution for a constant Ca^2+^ level is probably not hard. I am not insisting that any new simulations be included in the paper, but it would help to discuss briefly whether the assumption of two simultaneous Ca^2+^ binding affects the release latency estimation.

Answered in the list of essential revisions above.

2) On page 10, the Authors describe their checks to ensure that the model parameter optimization procedure is not stuck at a local minimum. However, a more straightforward common check would involve repeating the fminsearch parameter optimization algorithm multiple times, while starting at different initial values of parameters. Have the Authors performed such a check? This would essentially quantify uncertainties in inferred parameter values in Table 2.

Answered in the list of essential revisions above.

3) Apart from the nice connection between the proposed model and the model of Lou, Scheuss and Schneggenburger (2005), I think the presented model can also be viewed as a more detailed and biophysically-based extension of the "excess-calcium binding site model" of S.D. Meriney and coworkers, which I would recommend citing. To my knowledge, there are 3 papers that make use of the latter model, in one form or another (but I would recommend that the Authors double-check these papers to see if all of them are relevant):Dittrich M, Pattillo JM, King JD, Cho S, Stiles JR, Meriney SD (2013) An excess-calcium binding site model predicts neurotransmitter release at the neuromuscular junction. Biophys J 104: 2751-63Ma J, Kelly L, Ingram J, Price TJ, Meriney SD, Dittrich M (2015) New insights into short-term synaptic facilitation at the frog neuromuscular junction. J Neurophysiol 113: 71-87Luo F, Dittrich M, Cho S, Stiles JR, Meriney SD (2015) Transmitter release is evoked with low probability predominately by calcium flux through single channel openings at the frog neuromuscular junction. J Neurophysiol 113: 2480-9

We agree with the reviewer the excess calcium-binding site model has many similarities to our model. In our discussion, we now include a discussion of the original article. The follow-up papers of the model focus on facilitation and the origin of the Ca^2+^ ions that induce fusion, which we did not consider as relevant to our study.

Reviewer #2 (Recommendations for the authors):1. The style and length of this paper resemble a thesis. The authors are encouraged to edit the manuscript to make it accessible to a wide eLife audience. Many figures have complicated subpanels that are difficult to understand, and the text is often so technical that it does not convey the essential point of each argument. This is a very nice model. The authors could do a better job of describing the important points, which they highlight most clearly at the beginning of the discussion.

We appreciate this suggestion and have extensively rewritten and simplified the text. We now put a stronger emphasis on the biological relevance and have moved more technical sections to the methods. We have also worked to simplify our figures.

2. The model depicted in Figure 1A suggests that Ca-bound C2B attaches to the vesicle, rather than the PM. However, Chang, Trimbuch and Rosenmund 2019 showed that the PIP2 binding attaches vesicles to the PM before an action potential. Do the authors want to claim that Ca-bound C2B attaches to vesicles? This is actually very important to the main point of the paper, because "crosslinks" are likely to occur in the absence of Ca, and Ca-independent PIP2 binding might position the Ca-binding pocket closer to negatively charged phospholipids, increasing the affinity of C2B for Ca.

We realized that our term “crosslink” and the illustration of our model did give the impression that the C2B domain must bind both the plasma membrane and the vesicular membrane to exert its action. However, this is only one possible mode of action and in fact our model does not rely on this assumption at all. We now tried to clarify this by directly pointing out that it is currently not known how the C2B exerts its effect on the fusion barrier and name several possible mechanisms (induction of curvature, switch in the local electrostatic environment, direct or indirect regulation of SNARE complex assembly and distance regulation). To clarify this have removed the term “crosslink” from our manuscript. We have also included and additional supplementary figure (Figure 1 —figure supplement 1) for another alternative that the C2B only associates to the plasma membrane.

3. In the discussion (Line 542) the authors note that their model is fundamentally different from the proposed preassembly of Syt rings at the base of a vesicle, and that this is big testable prediction. How could this be tested? And do Syts actually preassemble with slots in their model? Figure 3B shows the number of crosslinks that precede fusion, and it would be nice to see how many Syts are bound to PIP2 at rest for any given number of "slots".

Answered in the list of essential revisions above.

4. Line 149: What (little) is known about the concentration of PIP2 at active zones? How does the concentration of PIP2 in rich patches compare to the idea that only 3 PIP2 molecules could be available beneath the space of one vesicle? I found the discussion of PIP2 concentration in Methods (Line 999) difficult to understand.

Currently little is known about the exact density of PI(4,5)P_2_ at release sites. Indeed, we know that PI(4,5)P_2_ forms clusters on membranes, which contain a high density of PI(4,5)P_2_ (Honigmann et al., 2013; Milosevic et al., 2005; van den Bogaart et al., 2011a). Clustering of PI(4,5)P_2_ on the plasma membrane is one of the reasons why we implemented slots in our model, but there are also several other explanations why the number of syts simultaneously binding to PI(4,5)P_2_ is limited. The number of slots in our model, which is set to three, does not correspond to the number of PI(4,5)P_2_ molecules beneath the space of one vesicle. The slots in our model set a limit in the number of Syts that can bind to PI(4,5)P_2_ simultaneously. We assume that each slot contains a certain PI(4,5)P_2_ concentration, which allowed us to directly implement binding affinities determined in the in vitro experiment by van den Bogaart et al., to constrain our model. Importantly, as for all models with three or more slots, we observe that most fusion events occur when three dual-bounds are formed, the number of slots in our model will not impact the main conclusions we draw. We now extend on the discussion of these concepts and the simplifications that were necessary to implement the model.

The section on the interpretation of the PI(4,5)P_2_ concentration in Methods was indeed quite technical. The main point we wanted to make in this section is that our model leads to an estimation of the concentration of PI(4,5)P_2_, whereas an estimate of the density of PI(4,5)P_2_ in the membrane would be more relevant, as PI(4,5)P_2_ is not dissolved in the cytosol. However, to compute the density from the estimated concentration we would need to include many assumptions. We removed this section from Methods and implemented the points in the Discussion.

5. The model only accounts for Ca binding by the C2B domain, which as the authors note is the more important of the C2 domains. C2A is barely discussed in this paper. However, there are differences in fusion rates when the C2A domain is mutated to block Ca binding. How would the addition of C2A affect the model?

We agree with the reviewer that we should discuss the C2A domain of Synaptotagmin more in our paper. We therefore included an additional section in the discussion.

The number of Ca^2+^ ions needed to bind to Synaptotagmin to induce fusion is the only parameter in our model that is based specifically on the C2B domain. The used Ca^2+^ and PI(4,5)P_2_ affinities are obtained using a construct containing both the C2A and C2B domain. However as mutating the Ca^2+^ binding sites of the C2A domain did not significantly affect the measured affinities, those affinity values can be attributed to the C2B domain (van den Bogaart, et al., 2012). Moreover, as we fitted our model to experimental data which is obtained in the presence of both the C2A and C2B domain of Synaptotagmin, our fitted model parameters likely also include some properties of the C2A domain. Therefore, although we focus on the C2B domain of Synaptotagmin, properties of the C2A domain are most likely indirectly included in our model simulations. For instance, quantification of the Ca^2+^ sensitivity of dense core vesicle fusion revealed a change upon mutation of the C2A domain binding sites (Soerensen et al., 2003), indicating that Ca^2+^ binding to the C2A domain may affect the affinity of the C2B domain. A similar effect in our model is attributed to a change in the PI(4,5)P_2_ concentration. Thus, a contribution of the C2A domain may be indirectly represented by our determined parameters. We agree with the reviewer that the initial version of the manuscript contained too little discussion on the C2A domain. We therefore now wrote a section on the C2A domain in the discussion.

An explicit simulation of the role of the C2A domain would require the inclusion of three more Ca^2+^ binding pockets with other, unknown Ca^2+^ affinities. And it might require additional assumptions on how the energy barrier for synaptic vesicle fusion is affected if Ca^2+^ ions bind to the C2A domain. While this may indeed be relevant for synaptic function, our simpler model focussing on the C2B domain only was sufficient to account for the experimental data we evaluated in this study. We thus chose not to explicitly include Ca^2+^ binding to the C2A domain in the context of this model.

6. I like Figure 2 Sup2, where the authors probe the number of crosslinks prior to fusion. Interestingly, the speed of Ca ramping affects the number of crosslinks, and a significant number of events occur after 4 crosslinks only when Ca increases at the fastest rate (0.001 s). This is still much slower than the expected rate of Ca nanodomain formation (<100 us). Why did the authors explore only extremely slow Ca ramp rates? It seems that the ramping rates of 10 us to 10 ms would more accurately affect the condition of action potential induced Ca influx.

Answered in the list of essential revisions above.

7. It is not immediately clear to me why the model with 6 slots never predicts fusion with 5-6 crosslinks. Why does the model fail to fit well if each crosslink contributes less to lowering the energy barrier, and thus 5-6 crosslinks are required to drive fusion?

Answered in the list of essential revisions above.

8. If a single crosslink brings the vesicle closer to the PM, this will introduce a new form of allostery by increasing the effective concentration of PIP2 sensed by Syts. Is this reflected in the model?

Answered in the list of essential revisions above.

9. In the model, PIP2 binding is required for Syt1-driven fusion. As a result, only Syts with a "slot" can participate in fusion. However, while K325, 327A mutations that disrupt Ca-independent PIP2 binding lead to less synchronous release, synchronous release can be restored using a paired-pulse stimulation paradigm where an initial action potential drives Ca-dependent membrane attachment of vesicles to the plasma membrane. This suggests that PIP2 binding is not required for "cross linking". How might the model change if Syts without slots could also contribute to fusion in your model?

Answered in the list of essential revisions above.

10. The evaluation of mutants in Figure 5 should be better tied to actual biology. What mutation does "Ca^2+^ binding", and "A-on" mimic? No citations are provided for the many studies where Syt mutants were expressed in Syt1 KO neurons. This seems like a perfect place to test the role of PIP2 binding with more complicated Ca stimuli. Could the model be adapted to explore the effect of K/A mutations described by Chang, Trimbuch and Rosenmund?

Answered in the list of essential revisions above.

Reviewer #3 (Recommendations for the authors):Such kind of complex models tend to have various underlying assumptions which could significantly influence the conclusions. Three examples are provided:

All models are based on assumptions and rested on observations. In all assumptions we made we aimed to match currently available experimental evidence as closely as possible. Additionally, we tried to keep the model as simple as possible to limit the number of assumptions needed to be made. To add strength to the model predictions performance of experiments would be of great value. However, some of the model predictions, such as the importance of the allosteric interaction for the function of Synaptotagmin, currently cannot be tested experimentally. This is often the motivation of deriving a model which can make testable predictions. We therefore believe that our modelling approach provides unique insights in the working mechanism of synaptotagmins.

1. The interaction of synaptotagmins with SNARES or other proteins could change the Ca^2+^ affinity but only PIP2 is considered as a potential interaction for changing the Ca^2+^ affinity. In other word, the premise of the paper that the Ca^2+^ affinity of synaptotagmin in vitro and in vivo (within the protein complex of the fusion machinery) is identical could be wrong. Therefore, the main conclusion of the strong cooperativity of Ca^2+^ and PIP2 could be wrong.

We respectfully point out that models are not typically “judged” on whether all assumptions and predictions are going turn out to be “right” in the future, rather by their use to make a synthesis of current knowledge and to make predictions inaccessible to measurements. Clearly, with future insights also our model will need to be extended.

We agree with the reviewer that interactions with the SNARE complex could also potentially increase the Ca^2+^ affinity of synaptotagmin by binding to its polybasic sequence via a similar mechanisms as PI(4,5)P_2_ binding. However, such a situation would still be consistent with our model and the strong allosteric mechanism between the Ca^2+^ and PI(4,5)P_2_ and/or SNARE binding sites essential for the protein’s function (as we point out). Therefore, this point is not at odds with our model and the Ca^2+^ affinity in vitro and vivo would match, simply the interpretation of the molecular interaction responsible for the Ca^2+^ affinity increase differs.

2. The assumption that each crosslinking synaptotagmin lowers the energy barrier for fusion by the same amount (E_syt) could be wrong. Assuming a positive or negative cooperativity in E_syt per synaptotagmin might impact the conclusions regarding the number of required synaptotagmins.

It is a possibility that the additive contribution of individual syts to the energy barrier is non-linear. However, even if this were the case, there is currently no experimental measurement that would allow to constrain such a complex scenario, even the direction of the cooperativity (negative or positive) cannot be predicted. Implementation of such further mechanisms to the model would go against this reviewer’s primary concern as it would require additional assumptions and unknown parameters. We avoid this uncertainty by investigating the simplest scenario (independent and additive effects).

In case of a positive cooperativity, i.e. each dual-bound syt having an increasing effect on the effect on the height of the energy barrier for synaptic vesicle fusion, our model would still predict the involvement of at least two synaptotagmins in fusion, as we could not fit our model with only 1 slot (Figure 2). With a negative cooperativity, i.e. each dual-bound synaptotagmin has a reducing effect on the reduction of the height of the energy barrier, the number of synaptotagmins actively involved in fusion could be larger than three. Similar to the points raised above, such specialisations would infer slight modifications of the model, but would not lead to a different model per se.

3. The distance between the individual synaptotagmin molecules and the nearest Ca^2+^ channels could differ significantly on the nm-scale. The assumption of an identical local Ca^2+^ signal for all synaptotagmins could be wrong which could complicate the model predictions of the effect of reduced synaptotagmin copy numbers (Figure 4 and 5).

The data the model was fit to were obtained in Ca^2+^ uncaging experiments. In these experiments, the distance to voltage gated Ca^2+^ channels does not matter because the stimulus is spatially homogenous. Also, to simulate the action potential evoked EPSCs, we used a previously published calcium transient that was calculated from calcium-uncaging data and measured EPSC waveforms, while assuming that all vesicles experience the same Ca^2+^-signal (Wang, Neher, Taschenberger 2008). This can be viewed as the ‘average’ Ca^2+^ signal RRP vesicles experience upon arrival of an action potential. We agree that there can be interesting effects of combining a heterogeneous vesicle-synaptotagmin number with heterogeneous vesicle-channel distances, but this will again require additional unknown parameters.

In conclusion, the study provides interesting and plausible possibilities of how synaptotagmins could mediate vesicle fusion but experimental validations are required to increase the amount of reliable and novel insights.

We thank the reviewer for the encouraging remarks and in our manuscript now point out several experiments that could be used to test different aspects of the model.